# TIR-Bench: A Comprehensive Benchmark for Agentic Thinking-with-Images Reasoning

## Abstract

The frontier of visual reasoning is shifting toward models like OpenAI o3, which can intelligently create and operate tools to transform images for problem-solving, also known as thinking-*with*-images in chain-of-thought. Yet existing benchmarks fail to fully capture this advanced capability. Even Visual Search, the most common benchmark for current thinking-*with*-images methods, tests only basic operations such as localization and cropping, offering little insight into more complex, dynamic, and tool-dependent reasoning. We introduce **TIR-Bench**, a comprehensive benchmark for evaluating agentic thinking-with-images across 13 diverse tasks, each requiring novel tool use for image processing and manipulation in chain-of-thought. We evaluate 22 multimodal large language models (MLLMs), from leading open-sourced and proprietary models to those with explicit tool-use augmentation. Results show that TIR-Bench is universally challenging, and strong performance requires genuine thinking-with-images capabilities. Finally, we present a pilot study comparing direct versus agentic fine-tuning.

## 1 Introduction

The reasoning abilities of recent multimodal large language models (MLLMs) (Hurst & OpenAI, 2024; Gemini Team, 2025) have advanced significantly, driven in large part by reasoning techniques such as chain-of-thought (CoT) (Wei et al., 2022). By decomposing reasoning of complex visual questions into a series of textual steps, MLLMs are able to achieve improved performance. While promising, these techniques are confined to the textual domain, conducting their reasoning solely through language while treating the visual information as a static, unalterable input (Su et al., 2025b).

To effectively process visual information, thinking-with-images has been proposed (OpenAI, 2025c; Su et al., 2025b; Hu et al., 2024). This approach enables a model to generate new visual information by actively manipulating input images with tools. For example, when faced with a complex visual problem, OpenAI's o3 model (OpenAI, 2025c) first writes code to create an image-processing tool, then executes it to modify the image (*e.g.*, cropping, flipping, or rotating). The transformed visual data then informs the next stage of its linguistic reasoning.

To assess the agentic thinking-with-images capabilities of MLLMs, existing benchmarks (Wu & Xie, 2024; Wang et al., 2025) have largely centered on visual search and tasks requiring the analysis of high-resolution images. These evaluations primarily validate a model's ability to accurately localize and crop specific regions within an image for better capturing detailed information to answer corresponding questions. However, these assessments tend to focus narrowly on the visual search capabilities of agentic MLLMs, leaving a broader spectrum of thinking-with-images abilities unevaluated. Therefore, there is an urgent need for a benchmark that incorporates a diverse range of tasks requiring sophisticated tool use to properly assess integrated multimodal reasoning.

In this paper, we introduce TIR-Bench, a comprehensive benchmark designed to evaluate the diverse thinking-with-images capabilities of agentic MLLMs. Unlike previous benchmarks focusing solely on visual search problems, TIR-Bench incorporates a diverse set of 13 tasks that require a wide range of tool-based interactions, such as zooming, rotating, increasing image contrast, adding auxiliary lines, and others to assess a model's tool integrated reasoning capabilities. The design of each task is predicated on the human intuition that solving it requires actively manipulating the visual input, rather than relying on static observation alone. For example, in TIR-Bench, a math problem might require the model to draw auxiliary lines or a coordinate system to find a solution, while a jigsaw

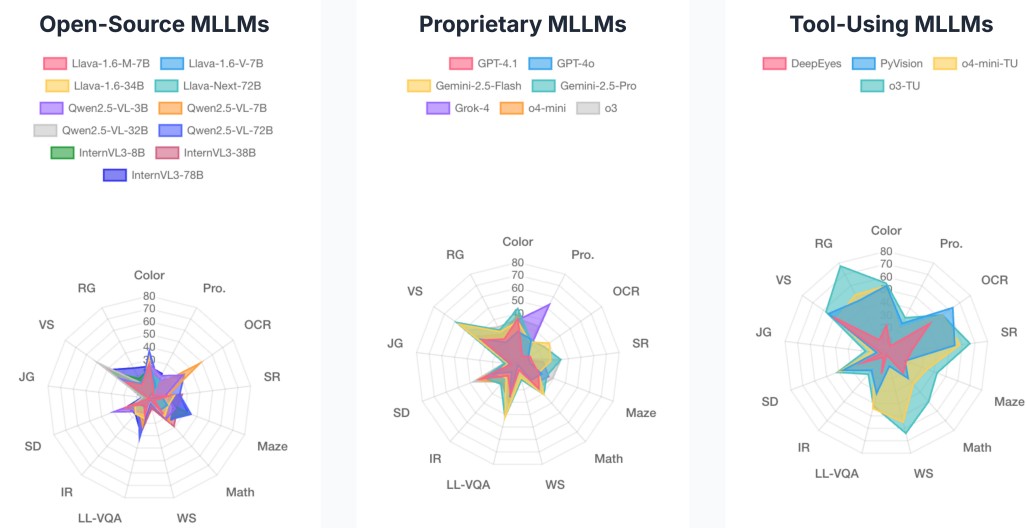

Figure 1: Overview of performance of open models (left), proprietary models (middle), and agentic tool-using models (right). SR: Symbolic Reasoning, WS: Word Search, LL-VQA: Low-Light VQA, IR: Instrument Reasoning, SD: Spot Difference, JG: Jigsaw Game, VS: Visual Search, RG: Rotation Game, Pro.: Proportion VQA. o3-TU: o3-tool-using, i.e., o3 with code interpreter.

puzzle task demands that it segment and then reassemble the image pieces. Consequently, TIR-Bench enables a more holistic evaluation of a model's thinking-with-images abilities, assessing a spectrum of skills not limited to visual search.

Using TIR-Bench, we conduct a comprehensive performance evaluation of 22 leading MLLMs across three categories: open-source models, proprietary models, and tool-using agents. The overall experimental results, illustrated in Figure 1, reveal that TIR-Bench is a challenging benchmark for thinking-with-images abilities, as the best performance achieved is only 46%. Moreover, traditional non-agentic models perform poorly on TIR-Bench, with the best-performing model Gemini-2.5-pro reaching an accuracy of merely 28.9%. These findings highlight the importance of the thinking-with-images ability for this benchmark, as models equipped with tool-use capabilities, such as o3, o4-mini, and PyVision (Zhao et al., 2025), achieve much higher performance than other models.

Lastly, we assess the function-calling proficiency of various MLLMs and conduct a pilot study contrasting direct supervised fine-tuning (SFT) with an agentic SFT approach on rotated image OCR task. Many recent MLLMs are equipped with function-calling capabilities. Our evaluation on rotation game task of TIR-Bench measures a model's proficiency in accurately executing the correct tool parameters as part of its reasoning chain. Results show that recent models like o3 perform well, whereas earlier models such as GPT-4o perform significantly worse. The pilot study on the rotated image OCR task compares two training methodologies and examines whether end-to-end SFT can achieve strong performance across different data scales for tasks involving image operations. Our findings indicate that the agentic SFT on full problem-solving trajectories with generated images is significantly more effective than direct SFT on rotated OCR task. This implies that agentic fine-tuning enables the emergence of more complex and robust problem-solving behaviors, allowing models to tackle multi-step tasks that are intractable with direct fine-tuning alone.

## 2 RELATED WORKS

### 2.1 MULTIMODAL BENCHMARKS

As the capabilities of Multimodal Large Language Models (MLLMs) evolve rapidly, a variety of benchmarks have been proposed to evaluate their performance, identify limitations, and guide future improvements (Lu et al., 2023; Zhang et al., 2024; Qiao et al., 2024; Wang et al., 2024a;b; Li et al., 2024c; Lu et al., 2021; Li et al., 2025). These benchmarks are typically either specialized for specific domains (Wang et al., 2024b; Li et al., 2024c; Lu et al., 2021; Li et al., 2025) or designed to be versatile and cover a broad range of tasks (Li et al., 2024b; Liu et al., 2023; Yue et al.,

2024; Wang et al., 2023). MLLMs often use CoT reasoning on these benchmarks, though solving them only requires static image information. More recently, benchmarks such as V* Bench and HR-Bench have been introduced to evaluate agentic capabilities, specifically for visual search in high-resolution images (Wu & Xie, 2024; Wang et al., 2025). While these benchmarks advance agentic evaluation by requiring models to programmatically crop high-resolution images, their focus is narrowly confined to visual search. Consequently, a broader spectrum of thinking-with-images abilities, such as rotation and drawing, remains largely underexplored.

## 2.2 THINK WITH IMAGES

In previous works, Visual Sketchpad (Hu et al., 2024), CoGCoM (Qi et al., 2025), DeepEyes (Zheng et al., 2025), Pixel Reasoner (Su et al., 2025a), OpenThinkIMG (Su et al., 2025b), Chain-of-Focus (Zhang et al., 2025a), Mini-o3 (Lai et al., 2025) and REVPT (Zhou et al., 2025) generate and executes tool calling in a predefined visual-specific toolset. In the intermediate reasoning steps, processed images are reinjected to the context, resulting in a multi-modal rational. However, these methods rely on a fixed collection of external visual parsers—such as detection models (e.g., GroundingDINO (Liu et al., 2024b)) and segmentation models (e.g., SAM (Kirillov et al., 2023a))—which constrains their generality across diverse vision tasks and introduces bottlenecks due to dependency on external models. In contrast, ViperGPT (Surís et al., 2023), o3 (OpenAI, 2025a), o4-mini, Thyme (Zhang et al., 2025b) and PyVision (Zhao et al., 2025) adopt Python as its primitive tool. Capitalizing on the advanced coding and multimodal understanding capabilities of modern MLLMs—such as Claude-4.0 (Anthropic, 2025) and GPT-4.1 (OpenAI, 2025b), enables the MLLM to dynamically write and execute code to construct complex, task-specific tools on demand, thereby supporting more general and flexible reasoning. This aligns with the emerging paradigm of "thinking with images" highlighted in o3's blog (OpenAI, 2025c) as a powerful cognitive capability. To assess this important ability, we propose TIR-Bench, covering diverse tasks on which multi-modal rationals are necessary.

## 3 TIR-BENCH

In this section, we introduce **TIR-Bench**. The overview of the benchmark is shown in Figure 2. We first introduce the task design strategy in subsection 3.1. Next, we introduce the data collection process in subsection 3.2. Finally, we present the benchmark summary in subsection 3.3.

## 3.1 TASK DESIGN

To extensively validate the model's ability to think with images we design 13 tasks. The benchmark's tasks are designed to evaluate a model's ability to perform active, tool-based visual reasoning, moving far beyond static image analysis. This includes tasks that require programmatic analysis, such as calculating color proportions or calling external models for object segmentation. Other challenges test the model's ability to overcome suboptimal conditions by programmatically enhancing low-light images or correcting the orientation of rotated text before performing OCR. The benchmark also features complex spatial and algorithmic puzzles, requiring models to solve mazes, reassemble jigsaw pieces, or draw auxiliary lines to solve geometric problems. Finally, it assesses fine-grained perception through tasks like spotting differences between images, reading instruments via cropping and zooming, and locating anomalies in visual puzzles. In every case, the model is forced to engage in a dynamic, multi-step process of visual manipulation and reasoning to arrive at the correct answer. More details about task design are introduced in Appendix B.

## 3.2 DATA COLLECTION

**Guidelines.** As previously mentioned, current benchmarks focus almost exclusively on visual search, overlooking a wide range of other tool-using abilities. To bridge this evaluation gap, we designed TIR-Bench in accordance with the following guidelines: (1) Diverse, Application-Grounded Tasks: TIR-Bench covers 13 distinct tasks across multiple domains, including spatial reasoning, visual perception, and mathematics, to mirror the complexity of real-world applications where static image analysis is insufficient. (2) Comprehensive Skill Assessment: The benchmark incorporates diverse visual contexts and requires a range of programmatic skills—from drawing auxiliary lines in

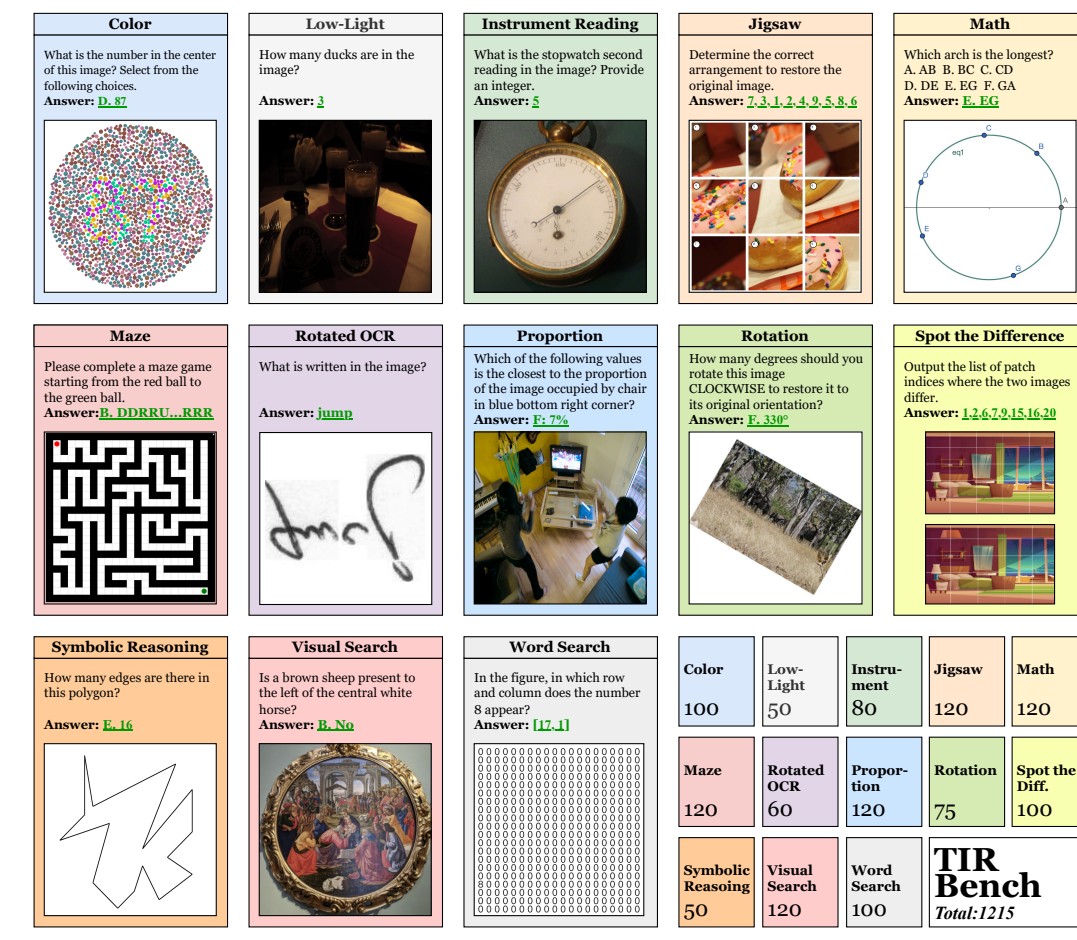

Figure 2: **Benchmark Overview.** TIR-Bench is composed of 13 tasks, meticulously designed to evaluate a wide spectrum of thinking-with-images capabilities.

geometry to executing pixel-level analysis—to foster a well-rounded evaluation of a model's agentic capabilities. (3) Probing Model Limitations: Each task is designed to be unsolvable without a multi-step, tool-based strategy. This intentional difficulty probes the limitations of current models, effectively distinguishing true thinking-with-images reasoning from simpler visual recognition. (4) Deterministic Evaluation: all tasks are designed with objectively verifiable answers, providing a robust framework for deterministic and reproducible evaluations. (5) Many samples in TIR-Bench are newly annotated or generated, making it a more reliable benchmark that minimizes the risk of data contamination from models' pre-training corpora.

**Collection.** We briefly introduce the data collection process of the 13 tasks here, while the detailed process can be found in Appendix B. The summary of data collection is shown in Table 1. Data from existing benchmarks are covered by the Apache License 2.0 or other licenses that permit academic use. For newly annotated data, images were sourced from freely available platforms like Wikimedia Commons and the National Gallery of Art, or were created by students using tools such as GeoGebra.

**(1). Collection of Math VQA, Symbolic Reasoning, Low-Light VQA, and Instrument Reading tasks:** Data from these tasks are newly created and annotated data. We tasked two Ph.D. students with sourcing free images from the Internet and MathVista (Lu et al., 2023) or create images using Geogebra, and then creating corresponding question-answer pairs. The annotators were explicitly instructed to design problems that, in their judgment, would necessitate tool-based image manipulation for a solution, ensuring the tasks could not be solved by static observation alone. Both students engaged in a reciprocal cross-checking process to verify the correctness of both the problems and their respective answers.

| Attribute | Color VQA | Proportion VQA | Symbolic Reasoning | Math | Word Search | Rotated OCR |
|---|---|---|---|---|---|---|
| Image Source | ColorBench | RefCOCO | VlmsAreBlind + Web-sourced | Web-sourced | Web-sourced | OCRBench |
| QA Construction | Human-annotated | Synthetic | Human-annotated | Human-annotated | Synthetic | Dataset-provided |
| Samples | 100 | 120 | 50 | 120 | 100 | 60 |
| Answer Type | Single-choice | Single-choice | Single-choice | Single-choice | Open-ended | Open-ended |

| Attribute | Maze | Low-Light | Instrument Reading | Spot the Difference | Jigsaw | Visual Search | Rotation |
|---|---|---|---|---|---|---|---|
| Image Source | Synthetic | Web-sourced | Web-sourced | Web-sourced | Synthetic | Web-sourced | CVBench |
| QA Construction | Synthetic | Human-annotated | Human-annotated | Human-annotated | Synthetic | Synthetic | Synthetic |
| Samples | 120 | 50 | 80 | 100 | 120 | 120 | 75 |
| Answer Type | Single-choice | Open-ended | Open-ended | Open-ended | Open-ended | Single-choice | Single-choice |

Table 1: Overview of all tasks, grouped into reasoning-oriented (top) and perception-oriented (bottom), with their sources, QA constructions, sample sizes, and answer types.

(2). **Collection of Color VQA task:** We tasked a Ph.D. student with curating samples from the ColorBench (Liang et al., 2025), specifically selecting instances that could not be solved by static observation alone and therefore necessitate programmatic analysis to answer correctly. Finally, we select 100 problems from ColorBench (Liang et al., 2025).

(3). **Collection of Jigsaw, Maze, Rotation, Word Search Tasks:** The data for these tasks were programmatically generated to create controlled and scalable challenges. For the Jigsaw Puzzle, we selected 120 images from the RefCOCO dataset, chosen for their prominent objects. These images were then segmented into grids ranging from 3×3 to 6×6 and shuffled to create puzzles of varying difficulty. For the Maze task, we programmatically generated 100 mazes with sizes scaling from 5×5 to 62×62. For the Rotation Game, we utilized 75 images from CVBench (Tong et al., 2024), to which we applied random rotations. Three difficulty tiers were established based on the magnitude of the rotation angle (e.g., 5, 10, or 15 degrees). Finally, for the Word Search task, we programmatically generated 85 samples of different sizes and supplemented them with 15 complex puzzles sourced from the internet.

(4). **Collection of Spot the Difference:** We sourced pairs of nearly identical cartoon and real-world images from the internet, with one image in each pair containing subtle alterations. Both images were then segmented into an $m \times n$ grid of corresponding patches. Finally, two annotators reviewed the pairs to identify and label the specific patches that contained the differences. Both students engaged in a reciprocal cross-checking process to verify the correctness of difference patch numbers.

(5). **Collection of Proportion VQA:** For this task, we collected 120 images from the RefCOCO dataset and used the ground-truth segmentation masks to calculate the correct object proportions. The incorrect multiple-choice options were then generated by adding or subtracting eight percentage points from the true value.

(6). **Collection of Rotated Image OCR Task:** We selected 60 images from OCRBench and applied a rotation to each. The rotation probabilities were 25% for 90°, 25% for 270°, and 50% for 180°.

(7). **Collection of Visual Search:** Recognizing that problems in existing benchmarks like V* Bench (Wu & Xie, 2024) are often too simple for current models, we curated a more challenging dataset for this task. We began by selecting 32 difficult problems from HR-Bench (Wang et al., 2025). To supplement these, we collected 88 new samples, which include 25 high-resolution art images and 63 high-resolution real-world images sourced from the internet. For each of these new images, an annotator was tasked with generating a unique question-answer pair. In total, the Visual Search task comprises 120 samples, 88 of which are newly created for this benchmark.

### 3.3 BENCHMARK SUMMARY

TIR-Bench consists of a total of 1215 examples divided into 13 diverse but essential tasks. Questions in our benchmark are categorized into two types: multi-choice and free-form problems, counting 665 and 550, respectively. The average length for question text is 46.29 and the average length for

answer text is 3.41. The distribution of each task is shown in Figure 7. Image and question examples for each task are shown in Appendix C.

# 4 EXPERIMENTS

In this section, we detail our evaluation setup and results for **22** leading MLLMs, considering both models with and without agentic tool-use capabilities. Our goal is to show that MLLMs lacking the ability to think with images perform poorly on TIR-Bench. We describe the experimental setup in subsection 4.1, report the results in subsection 4.2, present experiments on function calling in subsection 4.4, and compare agentic SFT with end-to-end SFT in subsection 4.5.

## 4.1 EXPERIMENT SETUP

**Model selection.** We categorize the MLLMs we use into open-source, proprietary, and tool-using. For open-sourced, we evaluated 11 models across three widely used and up-to-date model families: DeepSeek-VL-2 Wu et al. (2024), Kwai Keye-VL Team et al. (2025), Phi-4-Multimodal Abouelenin et al. (2025), LLaVA (Li et al., 2024a; Liu et al., 2024a), Qwen2.5-VL (Bai et al., 2025), InternVL3 (Zhu et al., 2025), and Qwen3-VL Team (2025), ranging from 3B to 78B. Results from these models accurately reflect open-source MLLMs' performance on thinking-with-image reasoning tasks. For proprietary models, we selected 7 models across 3 model families: GPT (Hurst & OpenAI, 2024), Gemini-2.5 (Gemini Team, 2025), and Grok-4 (xAI, 2025). We test GPT series, including GPT-4.1, GPT-4o (Hurst & OpenAI, 2024), as well as o-series models (OpenAI, 2025a), including o3 and o4-mini. For these GPT models, we use Azure API for calling models w/o python interpreter or sandbox (OpenAI, 2025c). For agentic tool-using (TU) MLLMs, we evaluate three open-sourced frameworks: DeepEyes (Zheng et al., 2025), and PyVision (Zhao et al., 2025), and 2 proprietary models o4-mini-TU and o3-TU. We use the official OpenAI API and turn on code interpreter and set the container parameter as auto.

**Evaluation.** We follow previous works (Lu et al., 2023; Li et al., 2025) to first generate answers from models and subsequently using GPT-4o to extract the final answer from the answer content. For multiple-choice and short-form answers, we compare the extracted value directly against the ground-truth to calculate accuracy; for grounding type problems such as Jigsaw Game and Spot the Difference with list type answer, we calculate the intersection over union (IoU).

**Implementation details.** We conduct all evaluations in zero-shot manner for fair comparison and better generalization. For open models, all experiments are done on NVIDIA A100 GPUs. For proprietary models, we use the official API. More details can be found in the Appendix E.

## 4.2 EXPERIMENTAL RESULTS

Both average and task-wise accuracies are reported in Table 2. We discuss several findings below.

**Result 1: TIR-Bench is challenging for all model types.** The highest performance observed among all models is only 46%, a result that underscores the difficulty of the TIR-Bench. This benchmark proves to be a significant challenge for assessing Thinking-with-Images capabilities, even for advanced models such as o3-TU, which leverage a code interpreter.

**Result 2: Traditional non-agentic models perform poorly on TIR-Bench.** Across all tasks, non-tool-using MLLMs show poor performance: the performance of most open-source models is close to or slightly higher than the random guess performance while the top-performing MLLM, Gemini-2.5-pro, surpasses random guess results by only 15%. These results highlight that, without agentic tool-using abilities, MLLMs can not perform well on TIR-Bench.

**Result 3: Agentic tool-using is essential for TIR-Bench.** The o3-TU model (OpenAI, 2025a) demonstrates the strongest overall performance, achieving the highest average accuracy at 46%. This represents a substantial lead, outperforming the Gemini-2.5-Pro model by nearly 17% and the o3 model without a code interpreter by 19%. With tool-use enabled, it achieves state-of-the-art

Table 2: Model Accuracy (%) Across Various Evaluation Tasks. SR: Symbolic Reasoning, WS: Word Search, LL-VQA: Low-Light VQA, IR: Instrument Reasoning, SD: Spot Difference, JG: Jigsaw Game, VS: Visual Search, RG: Rotation Game, Pro.: Proportion VQA. o3-TU: o3-tool-using, i.e., o3 with code interpreter. o3: o3 without code interpreter.

| Model | All | Color | Pro. | OCR | SR | Maze | Math | WS | LL-VQA | IR | SD | JG | VS | RG |
|---|---|---|---|---|---|---|---|---|---|---|---|---|---|---|
| Random Guess | 13.5 | 28.0 | 6.7 | - | 14.0 | 13.3 | 15.8 | 0.0 | 4.0 | 8.8 | 22.6 | 5.8 | 22.5 | 16.0 |
| *Open-Source MLLMs* | | | | | | | | | | | | | | |
| DeepSeek-VL2-Direct | 13.0 | 25.0 | 10.0 | 5.0 | 8.0 | 17.5 | 17.5 | 0.0 | 20.0 | 13.8 | 14.6 | 0.0 | 22.5 | 12.0 |
| DeepSeek-VL2-Think | 13.4 | 30.0 | 12.5 | 5.0 | 10.0 | 22.5 | 15.8 | 1.0 | 22.0 | 8.8 | 8.3 | 0.0 | 21.7 | 14.7 |
| Keye-VL-8B | 14.8 | 32.0 | 22.5 | 3.3 | 18.0 | 25.0 | 18.3 | 0.0 | 6.0 | 11.2 | 9.0 | 0.0 | 21.7 | 14.7 |
| Phi-4-Multimodal | 16.3 | 28.0 | 22.5 | 8.3 | 16.0 | 14.2 | 19.2 | 1.0 | 10.0 | 12.5 | 25.3 | 0.0 | 25.8 | 24.0 |
| Qwen3-VL-8B-Instruct | 16.9 | 30.0 | 10.0 | 41.7 | 14.0 | 30.8 | 18.3 | 2.0 | 20.0 | 16.2 | 9.8 | 0.0 | 22.5 | 14.7 |
| Qwen3-VL-8B-Thinking | 15.8 | 29.0 | 20.8 | 33.3 | 16.0 | 20.0 | 20.8 | 0.0 | 10.0 | 13.8 | 1.5 | 0.0 | 25.8 | 16.0 |
| Llava-1.6-M-7B | 11.3 | 27.0 | 7.5 | 3.3 | 16.0 | 4.2 | 16.7 | 0.0 | 14.0 | 6.3 | 18.0 | 0.0 | 22.5 | 12.0 |
| Llava-1.6-V-7B | 11.5 | 27.0 | 10.8 | 0.0 | 10.0 | 15.0 | 12.5 | 1.0 | 8.0 | 7.5 | 12.2 | 0.0 | 24.2 | 13.3 |
| Llava-1.6-34B | 13.0 | 31.0 | 6.7 | 1.7 | 20.0 | 15.8 | 18.3 | 0.0 | 16.0 | 16.3 | 11.9 | 0.0 | 21.7 | 10.7 |
| Llava-Next-72B | 11.2 | 20.0 | 15.8 | 3.3 | 8.0 | 10.8 | 15.0 | 0.0 | 10.0 | 11.3 | 16.3 | 0.0 | 23.3 | 12.0 |
| Qwen2.5-VL-3B | 17.7 | 27.0 | 20.0 | 31.7 | 12.0 | 21.7 | 20.8 | 0.0 | 12.0 | 13.8 | 29.7 | 0.0 | 26.7 | 12.0 |
| Qwen2.5-VL-7B | 16.0 | 21.0 | 10.8 | 48.3 | 14.0 | 15.0 | 24.2 | 0.0 | 22.0 | 11.3 | 24.4 | 0.0 | 21.7 | 9.3 |
| Qwen2.5-VL-32B | 18.7 | 26.0 | 19.2 | 25.0 | 10.0 | 18.3 | 23.3 | 2.0 | 14.0 | 15.0 | 13.1 | 5.4 | 48.3 | 13.3 |
| Qwen2.5-VL-72B | 19.7 | 37.0 | 15.0 | 33.3 | 24.0 | 35.0 | 22.5 | 3.0 | 32.0 | 12.5 | 14.1 | 0.0 | 25.8 | 12.0 |
| InternVL3-8B | 16.9 | 23.0 | 11.7 | 0.0 | 6.0 | 33.3 | 21.7 | 2.0 | 22.0 | 8.8 | 16.6 | 4.5 | 36.7 | 17.3 |
| InternVL3-38B | 19.1 | 24.0 | 10.8 | 3.3 | 26.0 | 23.3 | 29.2 | 8.0 | 28.0 | 13.8 | 14.6 | 5.1 | 44.2 | 13.3 |
| InternVL3-78B | 21.4 | 25.0 | 21.7 | 3.3 | 24.0 | 32.5 | 23.3 | 8.0 | 28.0 | 16.3 | 18.9 | 5.8 | 39.2 | 26.7 |
| *Proprietary MLLMs* | | | | | | | | | | | | | | |
| GPT-4.1 | 18.8 | 36.0 | 7.5 | 11.7 | 12.0 | 17.5 | 25.0 | 4.0 | 24.0 | 11.3 | 30.9 | 5.1 | 34.2 | 22.7 |
| GPT-4o | 17.3 | 26.0 | 22.5 | 10.0 | 10.0 | 20.0 | 15.8 | 0.0 | 26.0 | 7.5 | 19.4 | 6.2 | 35.0 | 20.0 |
| Gemini-2.5-Flash | 25.2 | 34.0 | 20.0 | 30.0 | 26.0 | 17.5 | 30.8 | 10.0 | 42.0 | 13.8 | 18.5 | 8.0 | 55.8 | 29.3 |
| Gemini-2.5-Pro | 28.9 | 44.0 | 21.7 | 25.0 | 34.0 | 24.2 | 30.8 | 12.0 | 42.0 | 20.0 | 28.5 | 10.4 | 58.3 | 30.7 |
| Grok-4 | 22.5 | 35.0 | 53.3 | 6.7 | 20.0 | 25.8 | 19.2 | 2.00 | 22.0 | 12.5 | 27.0 | 10.0 | 25.8 | 18.7 |
| o4-mini | 21.2 | 39.0 | 17.5 | 8.3 | 12.0 | 13.3 | 21.7 | 5.0 | 30.0 | 18.8 | 33.0 | 8.0 | 39.2 | 26.7 |
| o3 | 26.9 | 36.0 | 34.2 | 8.3 | 34.0 | 29.2 | 24.2 | 4.0 | 28.0 | 17.5 | 37.2 | 10.8 | 47.5 | 33.3 |
| *Tool-Using MLLMs* | | | | | | | | | | | | | | |
| Qwen2.5-VL-72B-TU | 24.0 | 39.0 | 26.7 | 41.7 | 20.0 | 23.3 | 20.8 | 3.0 | 36.0 | 18.8 | 19.2 | 5.3 | 45.8 | 21.3 |
| Qwen3-VL-235B-TU | 24.9 | 48.0 | 20.0 | 48.3 | 20.0 | 22.5 | 22.5 | 11.0 | 32.0 | 17.5 | 22.4 | 7.3 | 43.3 | 25.3 |
| DeepEyes | 17.3 | 22.0 | 6.7 | 41.7 | 19.9 | 16.7 | 20.0 | 1.0 | 16.0 | 3.8 | 19.9 | 3.9 | 50.8 | 12.0 |
| PyVision | 31.8 | 53.0 | 26.7 | 63.3 | 54.0 | 15.8 | 25.8 | 10.0 | 32.0 | 17.5 | 36.4 | 7.6 | 55.0 | 46.7 |
| o4-mini-TU | 37.5 | 53.0 | 21.7 | 53.3 | 58.0 | 34.2 | 31.7 | 55.0 | 44.0 | 13.8 | 38.9 | 11.8 | 47.5 | 52.0 |
| o3-TU | 46.0 | 55.0 | 31.7 | 53.3 | 66.0 | 42.5 | 50.0 | 64.0 | 42.0 | 21.3 | 41.0 | 16.4 | 57.5 | 77.3 |

results on the majority of the tasks, winning 10 out of the total 14 categories. The o3-TU and o4-TU models demonstrate a large performance improvement on tasks involving straightforward image manipulation or processing, including the rotation game, rotated image OCR, and word search. Similar phenomenon appears in PyVision, which implements thinking-with-images abilities based on GPT-4.1. PyVision brings 13% accuracy improvement compared with GPT-4.1.

We also observe that, although o3-TU excels in most categories, the improvements are not uniform. For example, for complex tasks such as *Jigsaw game*, the performance of o3-TU is still very low. On *Proportion VQA*, performance surprisingly decreases from 34.2% to 31.7%. This task requires calling external segmentation models such as Segment Anything (Kirillov et al., 2023b) to obtain a good rough estimate of the object segments. However, the current o3-TU model can only write executable code to manipulate images, but lacks the capability to call segmentation models specifically.

## 4.3 QUALITATIVE ANALYSIS

We present several examples of o3-TU responses here and examples of other model responses in Appendix. G. . Since the responses from the OpenAI API contain no reasoning process, we re-ran the questions in the web-based ChatGPT interface and analyzed the responses. Figure 4 shows an case of instrument reading. The model repeatedly crops the region around the pointer, checks its position, and guesses the answer (highlighted in blue). After several cycles of cropping, observing, and reflecting, o3 finally confirms its answer as "400," which is correct. Figure 5 shows a case of jigsaw puzzle involved a lot of ineffective attempts. The model first divided the complete image into segments, then tried to compare and judge similarity by "understanding" the image as a whole. In the early empirical stage, the model made nearly 55 iterations, but failed to obtain any usable partial results. After that, it began to use edge comparison, sampling the borders of image pieces

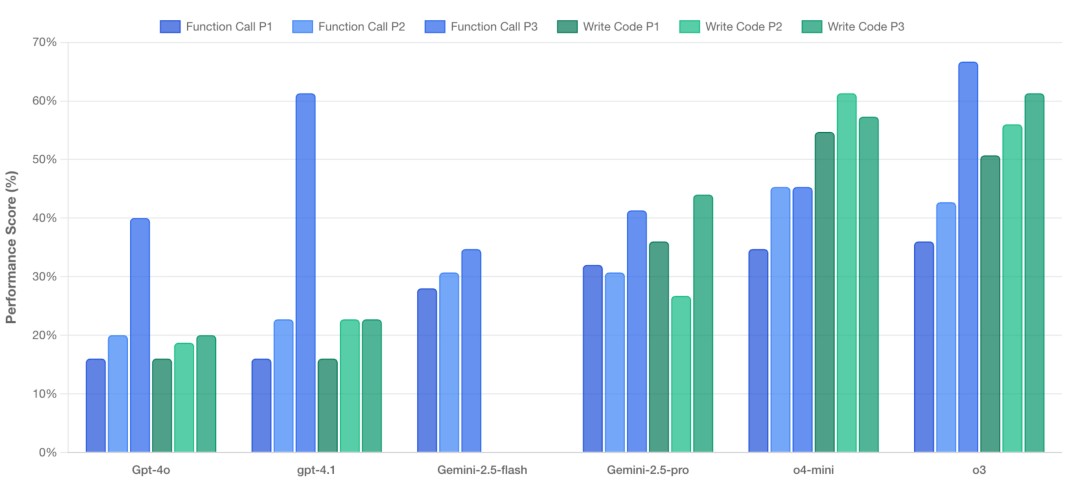

Figure 3: Comparison of function-calling and code-writing abilities across models. Function calling uses a predefined rotation function, while code writing requires the model to implement it. P1 includes only the question; P2 adds a hint to use the function; P3 explicitly instructs testing each degree in the answer choices. Stronger models (e.g., o3) perform better at code writing, whereas weaker ones (e.g., GPT-4o) perform better at function calling.

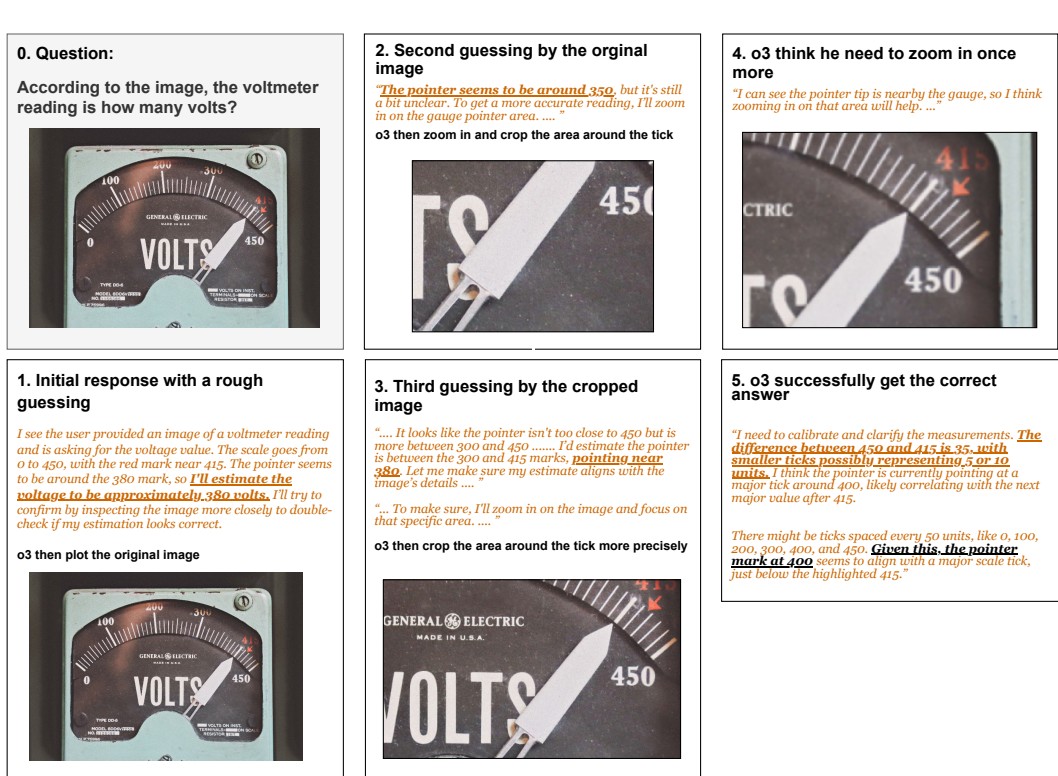

Figure 4: An example of o3 solving an case of instrument reading from TIR-Bench.

and comparing them pair by pair, which also did not produce effective outcomes. The model then started to develop algorithms, applying brute-force permutations to explore possible arrangements and calculate similarity. However, the initial algorithm did not work. After modifying the sampling method, the model, through more than 36,000 attempts, achieved a high similarity score that corresponded to the correct solution. This demonstrates that relying solely on the model's raw visual capabilities is insufficient; only code-based perception proves to be reliable.

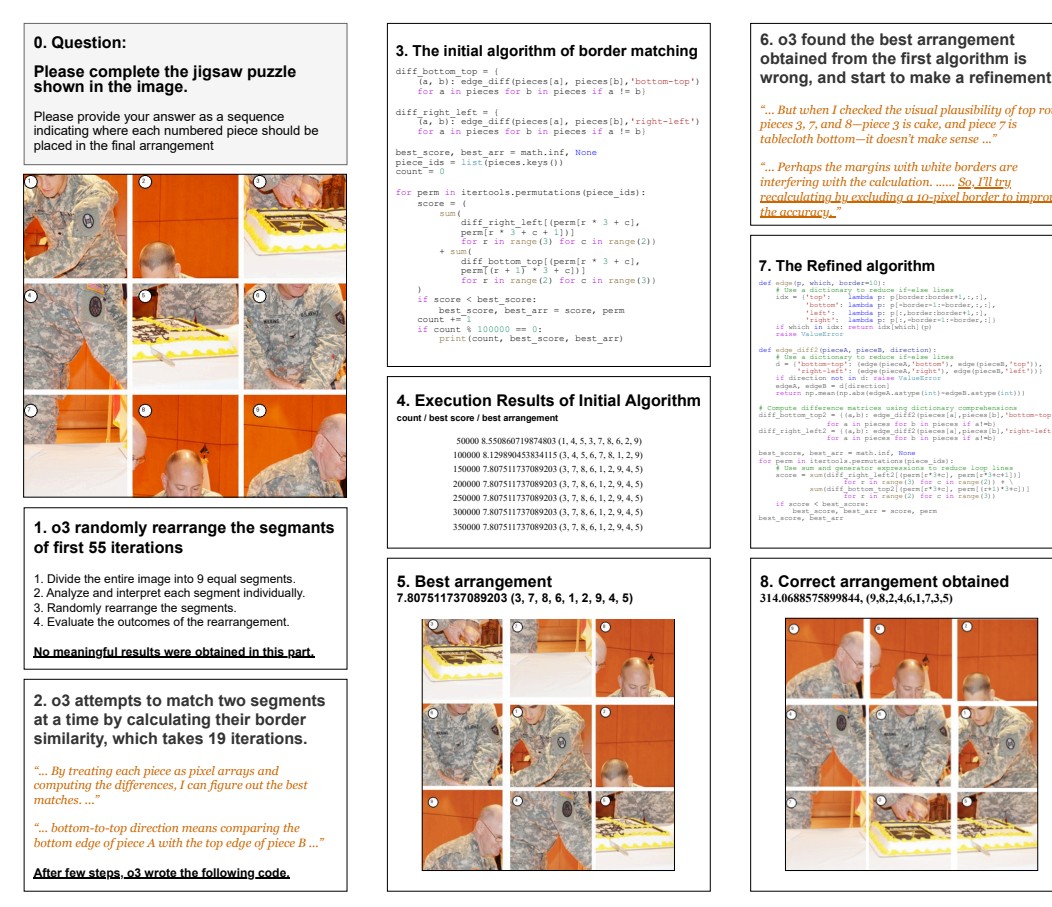

Figure 5: The o3 model's reasoning process for a jigsaw puzzle task in TIR-Bench involved a large number of ineffective attempts. To describe o3's reasoning process more clearly and concisely, we summarize its key behaviors as **TITLES** for each stage in the illustration. Important thought processes are highlighted in orange, and the **main code** implementations along with their corresponding output results are provided.

### 4.4 FUNCTION CALL EXPERIMENT RESULTS

In this subsection, we report the results on the function calling ability of different MLLMs.

**Set-up.** Using the Rotation Game task as a case study, we experimented with two distinct function-calling strategies: (1) providing the model with a predefined rotate function, requiring it only to output the degree parameter; and (2) requiring the model to generate the full image rotation code itself. Additionally, we tested three prompt variations to assess the impact of guidance: (1) a baseline prompt containing only the question (Prompt 1); (2) a prompt including a hint to leverage the rotation function (Prompt 2); and (3) a more explicit prompt instructing the model to systematically test each degree from the answer choices (Prompt 3). The specific function definitions and prompts used in this analysis are provided in Appendix D. During inference, we repeatedly call functions until the model produces a final answer without requiring any function parameters.

**Results.** We report the experimental results in Figure. 3 and report the average number of calling for each problem in Table. 3. Since Gemini-2.5-flash does not write code for all three prompts, we do not report it for writing code. We brief discuss some key findings here: **(1).** Clear Performance Hierarchy. o3 emerges as the top performer, achieving the highest accuracy, with o4-mini following closely. **(2).** Prompting Strategy is Key. The prompt strategy hinting to check each degree choice (prompt 3) works best. The performance of most models increases with prompt 3 compared to the other two prompts. This implies that models may not know how to best utilize the functions without explicit guidance. **(3).** Increased function call number in Newer Models. The average number of function calls for more recent advanced models (e.g., o3) is much higher than for previous models (e.g., GPT-4o). This implies that recent models are better trained for iterative function calling.

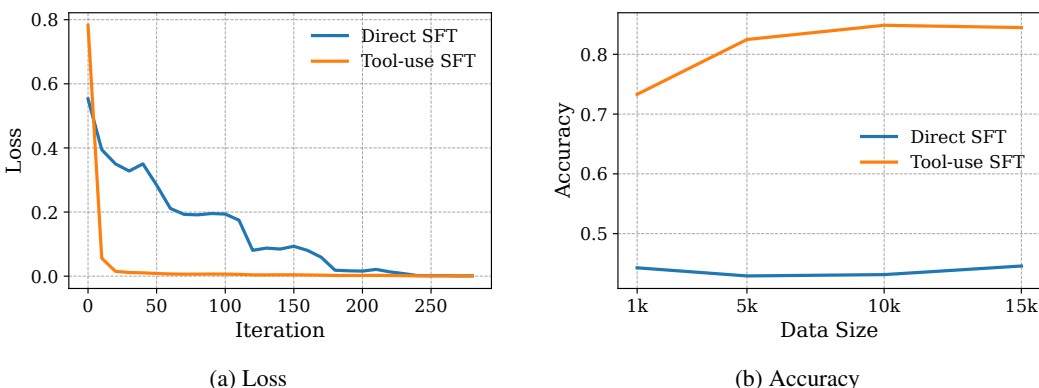

Figure 6: Comparison of change of loss and accuracy between direct SFT and tool-use SFT.

### 4.5 FINE-TUNING COMPARISON EXPERIMENT RESULTS

In this subsection, we report the experimental results on the different fine-tuning strategies on rotated OCR task.

**Setup.** We use the rotated image OCR task as a case study to evaluate data scaling performance. We created training sets of varying sizes: 1k, 5k, 10k, and 15k samples—by randomly selecting and rotating images from the OCRDataset (Minh, 2024). We then compared two distinct training strategies: (1). Direct SFT: A standard supervised fine-tuning approach where the model is trained to map the rotated image directly to the ground-truth text. (2). Tool-Use SFT: An agentic approach where the model first learns to output the correct rotation degree. The restored image is then concatenated with the original context, and the model is subsequently trained to read the text from this corrected visual input. We use Qwen-2.5-VL-7B and fully fine-tune all parameters with 5 epochs.

**Results.** We report accuracy on the Rotated OCR task in Figure 6b and the loss curves on 15k samples in Figure 6a. Overall, Tool-use SFT significantly outperforms Direct SFT. For Tool-use SFT, performance scales positively with data size, whereas Direct SFT shows no such trend. This suggests that simply scaling data for Direct SFT is ineffective on tasks requiring image-based reasoning. We also observe that Tool-use SFT's loss decreases much faster despite starting from a higher initial value. This is likely because Qwen-2.5-VL was not pretrained with function-calling, leading to higher initial loss. Furthermore, since Qwen-2.5-VL was trained mostly on correctly oriented OCR data, fine-tuning it directly on rotated data may cause forgetting. In contrast, restoring image orientation before text output avoids this issue, which may explain the faster loss reduction.

## 5 CONCLUSION

In this paper, we propose TIR-Bench, a comprehensive benchmark designed to evaluate the thinking-with-images ability of agentic MLLMs. TIR-Bench is composed of 13 meticulously collected tasks that assess a wide range of tool-assisted reasoning skills. By assessing diverse MLLMs on TIR-Bench, including both standard models and those augmented with tool-use capabilities, we find that TIR-Bench is a challenging benchmark for all models that necessitates thinking-with-images capabilities for successful completion. Lastly, we conduct a pilot study comparing direct and agentic fine-tuning for image-operation tasks and evaluating the function-calling abilities of various MLLMs.

## 6 LIMITATION

(1). The pilot study experiment comparing agentic SFT and directly SFT was conducted on only one task from our benchmark (Rotated-OCR). This task specifically tests an image manipulation-to-extraction pipeline. It remains an open question whether the superiority of agentic SFT would hold for the other 12 tasks in TIR-Bench. (2). We also note that some tasks, such as word search, are less common in real-world settings. We leave the inclusion of more real-world data for testing these abilities (e.g., pixel comparison) for future work.

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

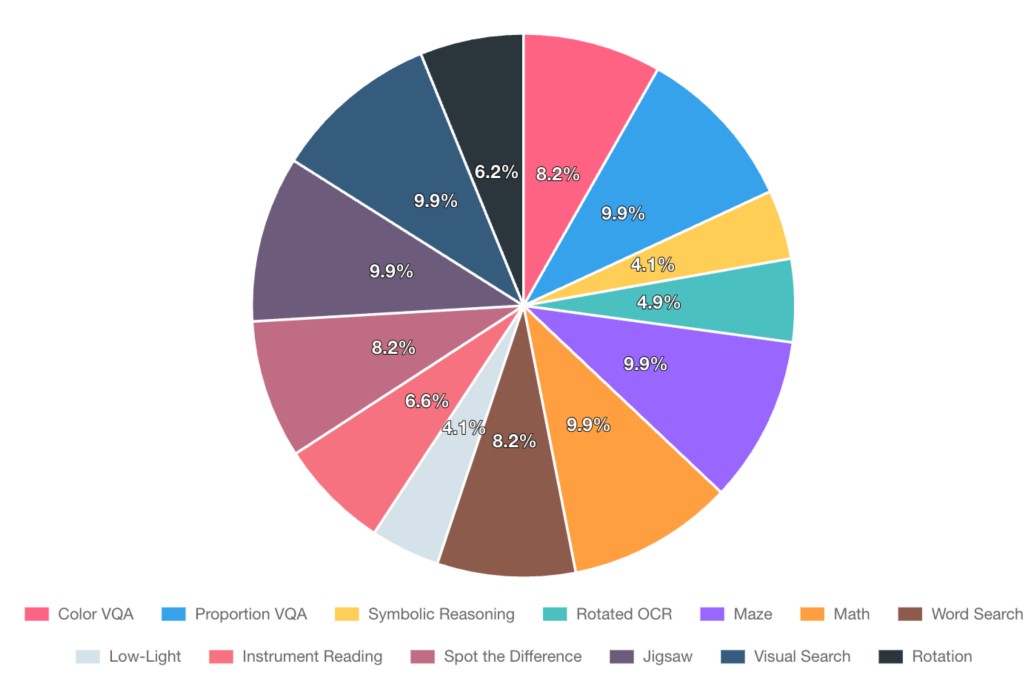

Figure 7: Task Distribution of TIR-Bench.

## A    THE USE OF LARGE LANGUAGE MODELS (LLMS)

In this paper, we only used LLMs to correct the grammar and spelling errors in the writing. All the results are produced by authors.

## B    DETAILED TASK DESIGN AND COLLECTION

### B.1    TASK DESIGN

To extensively validate the model's ability to think with images we design 13 tasks. The overview of the benchmark is shown in Figure 2. We describe these 13 tasks below:

- **Color VQA task:** This task assesses the model's ability to answer questions related to an image's color composition. Answering these questions requires the model to programmatically process the image to obtain visual information—for instance, by writing and executing code to calculate the proportion of a specific color. The data in this task are from ColorBench Liang et al. (2025)

- **Referring Object Proportion VQA**: This task assesses the model's agentic capabilities, requiring it to call a powerful external segmentation model to obtain an object's mask and then programmatically calculate its proportion relative to the entire image. The images are from RefCOCO Kazemzadeh et al. (2014)

- **Rotated Image OCR**: This task evaluates a model's ability to execute a multi-step visual reasoning process. The model must first identify that an image containing text is incorrectly oriented, then use a tool to rotate the image to its correct position, and finally perform optical character recognition (OCR) to accurately read the content. The images are from OCRBench Liu et al. (2024c).

- **Symbolic Reasoning**: This task assesses the model's ability to apply abstract, rule-based logic to visual information. For instance, when asked to count the edges of a complex polygon, the model cannot simply guess; it must systematically identify and enumerate each distinct edge, a process that may require internal algorithms for vertex or line detection to arrive at the correct count. The data on this task are from human-annotated or Rahmanzadehgervi et al. (2024)

- **Maze**: This task assesses the model's ability for advanced spatial planning and algorithmic execution. The model must analyze the visual structure of the maze, devise a solution using image processing tools (such as morphological operations), and apply a pathfinding algorithm to solve it. Finally, it must draw the solution path onto the image, demonstrating its ability to translate an abstract plan into a concrete visual action. Images and questions on this task are fully synthesized.

- **Math Problems**: This task assesses the model's ability to solve geometric problems by programmatically augmenting the visual input. To find a solution, the model must use tools to draw auxiliary constructs, such as adding lines to a diagram or imposing a coordinate system to define and calculate properties like relative lengths. Images and questions from this task are human-annotated or from MathVista Lu et al. (2023).

- **Word Search Puzzle**: This task evaluates the model's ability to perform fine-grained visual discrimination and anomaly detection. The image presents a field of numerous, nearly identical characters, with only a few subtle differences. Standard OCR is designed to fail in this scenario, forcing the model to devise a programmatic solution. To succeed, the model must write and execute code to perform a more fundamental analysis, such as a pixel-level comparison or a targeted visual search, to locate and identify the characters that deviate from the pattern. Images are synthesized or from free Internet platform (Wikipedia Common).

- **Low-Light Image VQA**: This task evaluates the model's ability to overcome suboptimal visual conditions. Given a low-light image where details are obscured, the model must first recognize the issue and then programmatically enhance the image, for instance, by writing and executing code to increase its contrast or brightness, before it can accurately answer questions about the content. Images are from Wei et al. (2018) and questions are annotated by humans.

- **Instrument Reading**: This task evaluates the model's ability to perform a sequential, tool-assisted analysis. To succeed, the model must first locate the key areas for reading the instrument, then programmatically crop and enlarge these specific regions for clarity, and finally, accurately read the value from the enhanced view. Images are from Wipipedia Common and https://unsplash.com/, questions are annotated by humans.

- **Spot the Difference**: This task evaluates the model's ability to perform precise, programmatic visual comparison. To identify the differences between two nearly identical images, the model must execute a tool-based strategy, such as programmatic image subtraction, to highlight differing regions at a pixel level. After locating the discrepancies, the model must then isolate and identify the specific image patches that contain these alterations. Images are from bought spot different puzzle books.

- **Jigsaw Puzzle**: This task assesses the model's ability to perform complex spatial reasoning through an iterative, tool-based approach. The model is required to programmatically segment an image into pieces and then repeatedly attempt to reassemble them. This process involves a continuous loop of action and self-correction, where the model must evaluate each configuration to determine if the image has been successfully restored, continuing the cycle until the solution is achieved. Images are from RefCOCO Kazemzadeh et al. (2014).

- **Visual Search**: This task evaluates the model's ability to locate specific targets within a complex or high-resolution image through deep, multi-turn reasoning. To succeed, the model must engage in an iterative search process, strategically using tools to repeatedly zoom in on and analyze different regions until the object or information is found. Data are from https://unsplash.com/, National Gallery of Art, and HRBench Wang et al. (2025).

- **Rotation Game**: This task assesses the model's ability to perform iterative orientation correction (Xie et al., 2025). To restore a rotated image to its upright position, the model must programmatically test various rotation angles. After each transformation, it must visually evaluate the result to determine if the orientation is correct, engaging in a trial-and-error loop of tool-based action and visual judgment until the problem is solved. Images are from CVBench Tong et al. (2024).

## C   DETAILED TASK EXAMPLE

### C.1   COLOR

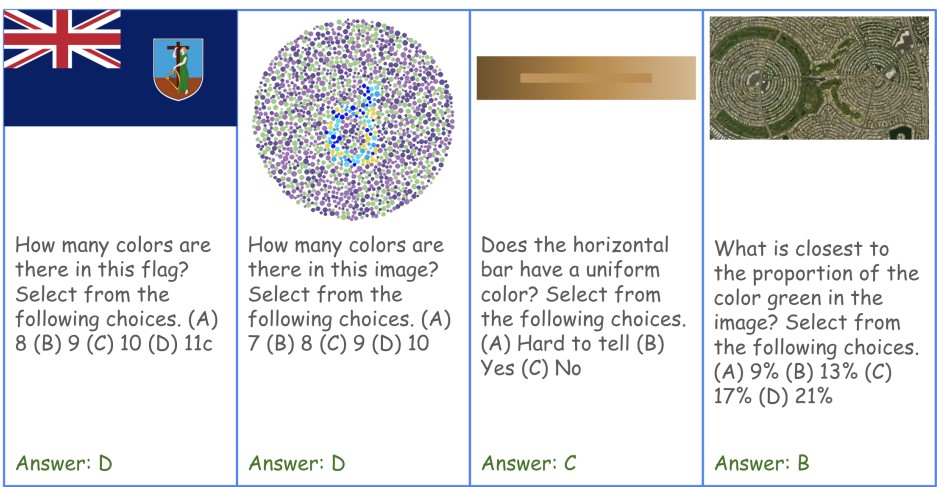

Figure 8: Additional example of "Color" task.

### C.2   LOW-LIGHT

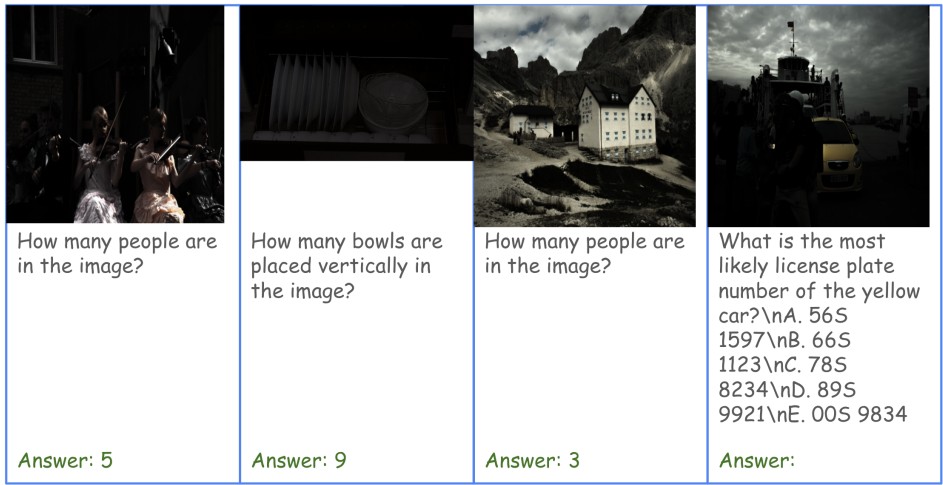

Figure 9: Additional example of "Low-Light" task.

## C.3 INSTRUMENT READING

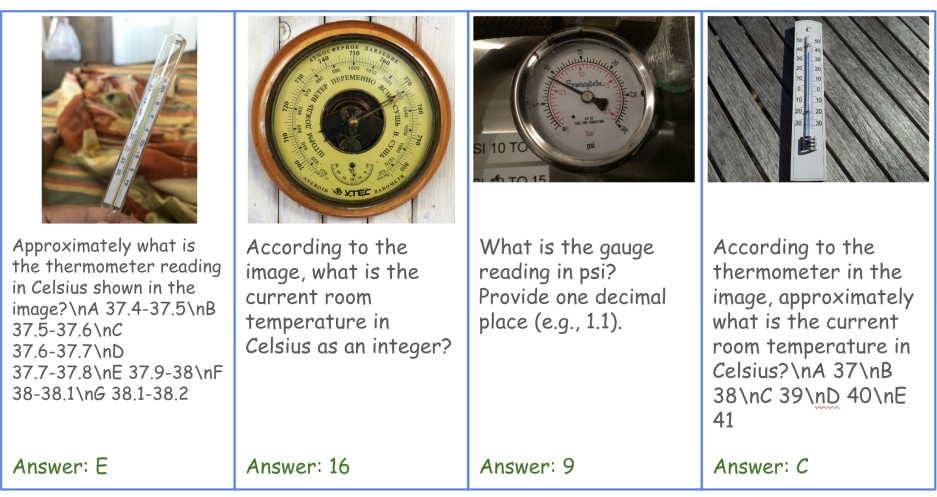

Figure 10: Additional example of "Instrument Reading" task.

## C.4 JIGSAW

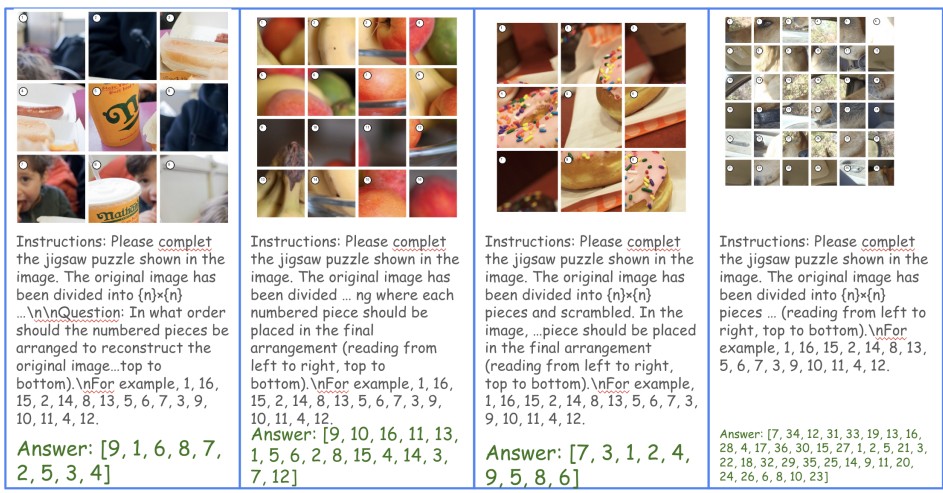

Figure 11: Additional example of "Jigsaw" task.

## C.5 MATH

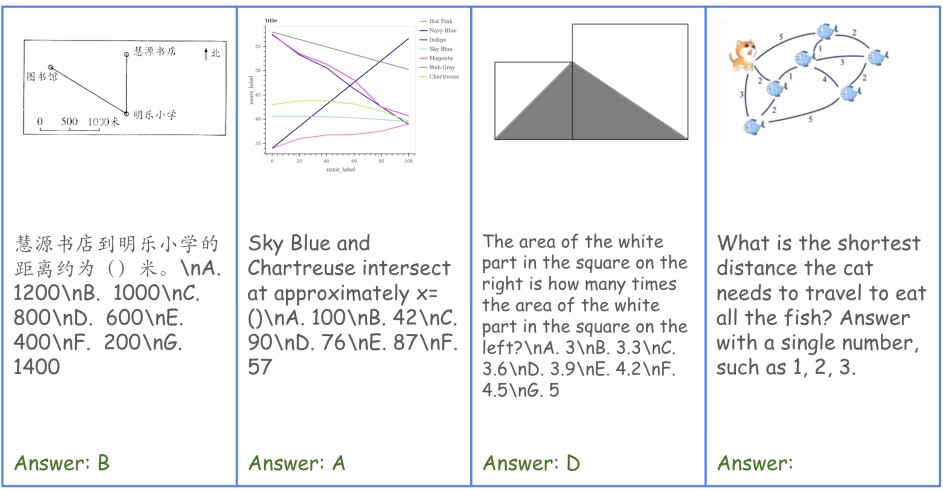

Figure 12: Additional example of "Math" task.

## C.6 MAZE

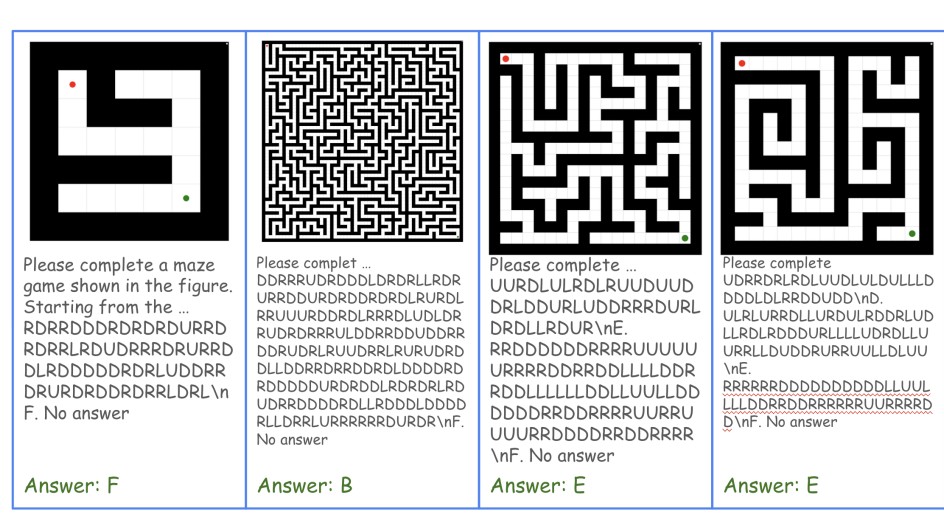

Figure 13: Additional example of "Maze" task.

### C.7    ROTATED OCR

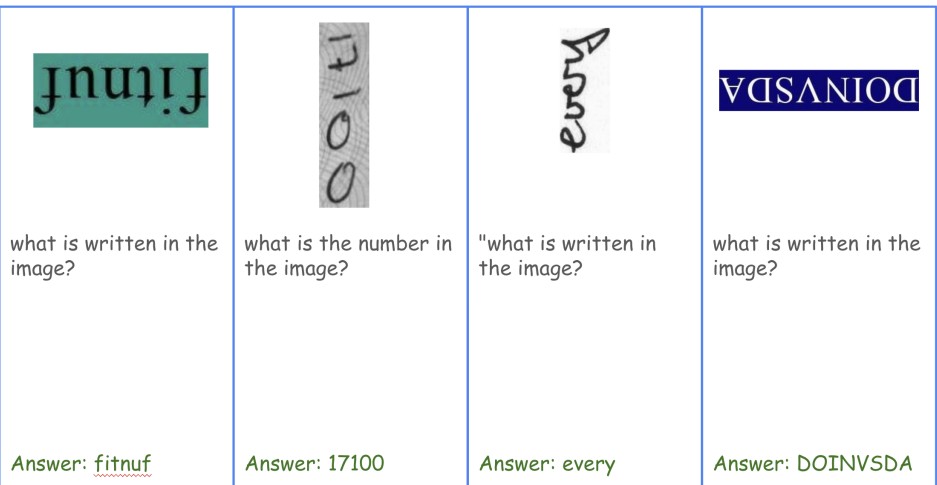

Figure 14: Additional example of "Rotated OCR" task.

### C.8    PROPORTION

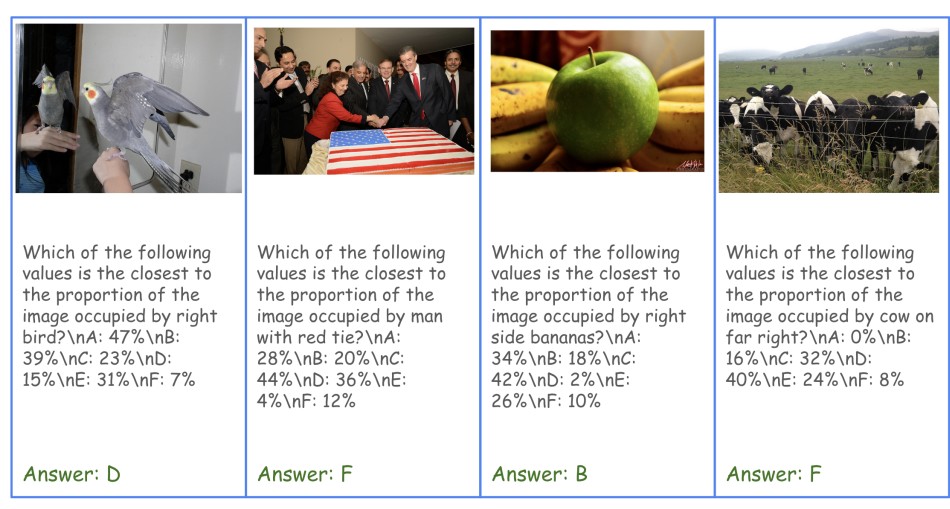

Figure 15: Additional example of "Proportion" task.

## C.9 ROTATION

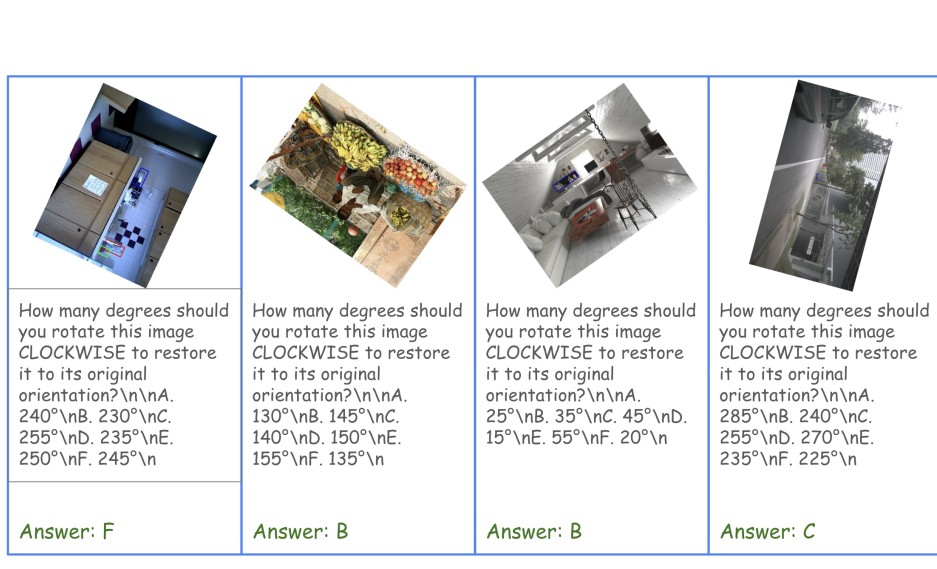

Figure 16: Additional example of "Rotation" task.

## C.10 SPOT THE DIFFERENCE

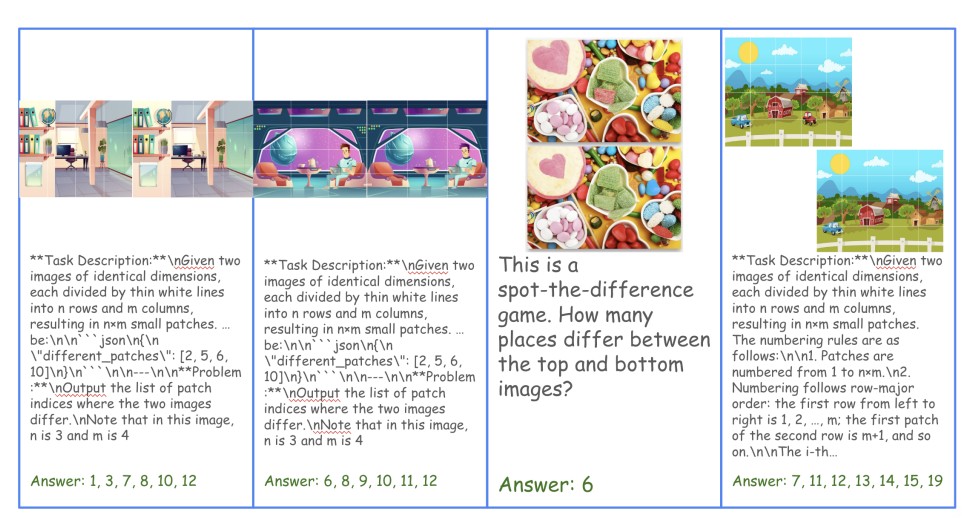

Figure 17: Additional example of "Spot the Difference" task.

## C.11 SYMBOLIC REASONING

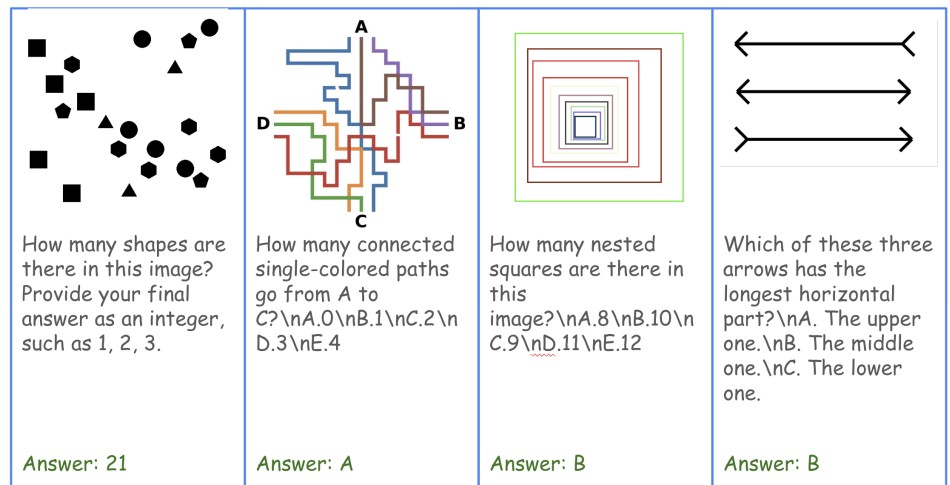

Figure 18: Additional example of "Symbolic Reasoning" task.

## C.12 VISUAL SEARCH

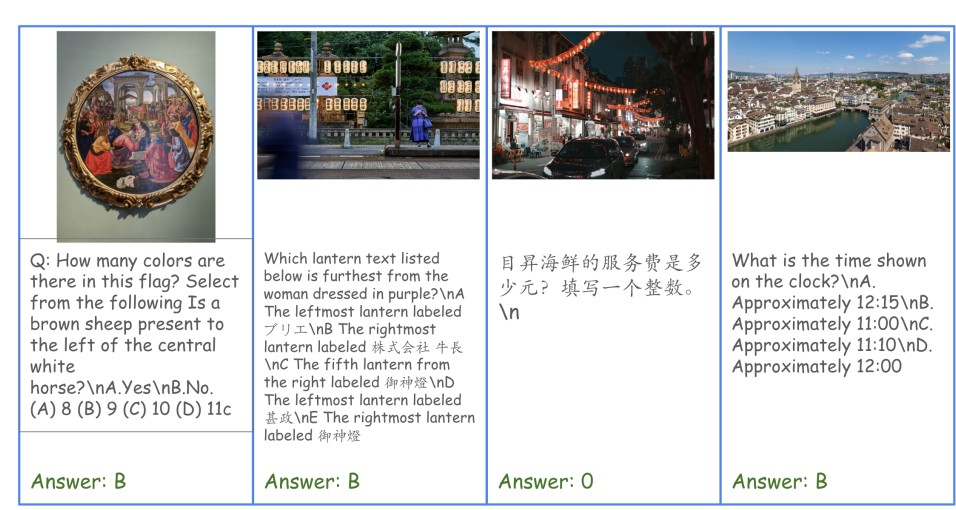

Figure 19: Additional example of "Visual Search" task.

## C.13 WORD SEARCH

| | | | |
|---|---|---|---|
| In the figure, in which row and column does the number 9 appear? The final answer should output two number list. The first number represents the row count from top to bottom, and the second number represents the column count from left to right. For example, [1, 2] means the first row and the second column | 图中有多少个正字？回答一个数字，例如1，2，3。 | 图中入字在出现在第几行第几列？最后答案输出两个数字，第一个数字代表从上往下多少行，第二个数字代表从左往右列。例如[1, 2]代表第一行第二列 | How many times does the number 9 appear in this image? Answer with an integer, such as 1,2,3. |
| Answer: [54, 50] | Answer: 9 | Answer: [31, 6] | Answer: 8 |

Figure 20: Additional example of "Word Search" task.

# D DETAILED PROMPTS

## D.1 WRITE CODE

prompt_being is the prompt for the beginning turn and prompt_return is the prompt for returning processed images to models.

Prompt strategy 1:

```
prompt_1 = user_prompt
prompt_2 = "The returning status and the processed image or text
(if any) of the code is attached."
```

Prompt strategy 2:

```
prompt_being = user_prompt + '\nPlease consider to write code to
process the image.'
prompt_return = 'The returning status and the processed image or
text (if any) of the code is attached, you can continue to write
code to process the original image for better understanding or
proceed to answer the question.'
```

For prompt 3:

```
prompt_1 = user_prompt + "\nPlease try to write code to rotate the
original image with the rotation degree in the options to verify
the correctness.  You can try as many as you can."

prompt_2 = "The returning status and the processed image or text
(if any) of the code is attached, you can continue to write code
to rotate the original image with a degree of other difference
options for better understanding or proceed to answer the question
if you have meet the correct option."
```

## D.2 FUNCTION CALLING

For prompt strategy 1:

```
prompt_1 = user_prompt
prompt_2 = "The rotated image is attached."
```

For prompt strategy 2:

```
prompt_1 = user_prompt + '\nPlease consider to call the rotation
function.'
prompt_2 = 'The rotated image is attached, you can continue to
call the function to rotate the original image with a difference
degree for better understanding or proceed to answer the
question.'
```

For prompt strategy 3:

```
prompt_1 = user_prompt + "\nPlease try to call the rotation
function with the rotation degree in the options to verify the
correctness.  You can try as many as you can."
prompt_2 = "The rotated image is attached, you can continue to
call the function to rotate the original image with a degree of
other difference options for better understanding or proceed to
answer the question if you have meet the correct option."
```

Table 3: Average number of function calling across different prompt strategies and calling stratefies.

| Model | Function | | | Write Code | | |
|---|---|---|---|---|---|---|
| | Prompt 1 | Prompt 2 | Prompt 3 | Prompt 1 | Prompt 2 | Prompt 3 |
| GPT-4.1 | 0.00 | 1.15 | 4.68 | 1.32 | 1.68 | 3.95 |
| GPT-4o | 0.00 | 1.12 | 4.29 | 2.15 | 1.87 | 4.07 |
| Gemini-2.5-flash | 0.64 | 1.05 | 1.73 | - | - | - |
| Gemini-2.5-pro | 0.75 | 0.87 | 1.09 | 0.25 | 0.28 | 0.21 |
| o4-mini | 0.07 | 1.52 | 2.87 | 3.43 | 3.68 | 2.68 |
| o3 | 0.33 | 1.57 | 3.59 | 6.05 | 5.77 | 5.76 |

## E    IMPLEMENTATION DETAILS

We conduct all evaluations in zero-shot manner for fair comparison and better generalization. For open models, all experiments are done on NVIDIA A100 GPUs. For proprietary models, we use the official API. We set do sample = False, temperature = 0, max new tokens = 2048 for all open models.

## F    MORE EXPERIMENTAL RESTULS

In this section, we discuss the results from synthetic data and human-annotated data. The results are shown in Table 4. First, models across all categories—from open-source to proprietary—consistently find Synthetic tasks more challenging than Annotated tasks. The average accuracy for Synthetic tasks (e.g., Maze, Jigsaw, Word Search) is significantly lower than for Annotated tasks (e.g., Color, Math, OCR). This suggests that current MLLMs struggle more with abstract spatial and logical reasoning than with recognition-based problems.

Second, the integration of tools provides a dramatic performance boost, especially for the difficult Synthetic tasks. The tool-using models, particularly o3-TU, dominate the benchmark. For example, o3-TU (43.1% Synthetic) achieves nearly double the synthetic-task accuracy of its non-tool-using counterpart, o3 (22.1%). This highlights that a tool-augmented architecture is critical for tackling these complex reasoning challenges.

Table 4: Model Accuracy (%) on Synthetic and Annotated data

| Model | Synthetic Acc. (%) | Hand-Annotated Acc. (%) |
|---|---|---|
| Random Guess | 8.0 | 18.2 |
| *Open-Source MLLMs* | | |
| Llava-1.6-M-7B | 4.3 | 16.8 |
| Llava-1.6-V-7B | 7.8 | 14.4 |
| Llava-1.6-34B | 6.5 | 18.1 |
| Llava-Next-72B | 7.6 | 15.0 |
| Qwen2.5-VL-3B | 11.0 | 22.9 |
| Qwen2.5-VL-7B | 7.1 | 23.0 |
| Qwen2.5-VL-32B | 11.9 | 24.1 |
| Qwen2.5-VL-72B | 13.5 | 24.6 |
| InternVL3-8B | 13.9 | 19.2 |
| InternVL3-38B | 12.2 | 24.5 |
| InternVL3-78B | 18.7 | 23.5 |
| *Proprietary MLLMs* | | |
| GPT-4.1 | 10.7 | 25.3 |
| GPT-4o | 13.7 | 20.1 |
| Gemini-2.5-Flash | 16.2 | 32.3 |
| Gemini-2.5-Pro | 19.2 | 36.5 |
| Grok-4 | 23.0 | 22.2 |
| o4-mini | 13.4 | 27.4 |
| o3 | 22.1 | 30.8 |
| *Tool-Using MLLMs* | | |
| DeepEyes | 8.0 | 25.4 |
| PyVision | 19.7 | 41.4 |
| o4-mini-TU | 32.8 | 41.3 |
| o3-TU | 43.1 | 48.2 |

## G    EXAMPLES OF MODEL RESPONSES

In this section, we provide the response of different models.

## H    ERROR ANALYSIS

We found that approximately 37% of PyVision's erroneous responses stem from perception errors, such as cropping incorrect image patches or misinterpreting visual content. For example, in visual search and instrument-reading tasks, PyVision may crop the wrong regions or misread instrument values. Another 61% arise from tool-generation and reasoning errors, including language-reasoning mistakes and flawed logic when generating code for image processing (e.g., producing tools that fail to solve the target problem). For instance, in jigsaw tasks, PyVision may incorrectly split the image into patches or rearrange the patches only once, resulting in wrong answers. The remaining 2

## I    SUMMARY OF PYVISION RESPONSE TOOL USAGE

Because the OpenAI API does not provide intermediate reasoning or code, we have summarized the Pyvision model's tool usage. We use GPT-4o to summarize and extract the tool of generated code. We describe these 13 tasks below:

- **Color VQA task:** pixel color counting, pixel searching, color thresholding, crop, drawing color channel histogram, image enhancement, converts an image to grayscale, color proportion calculation, pixel color analysis.

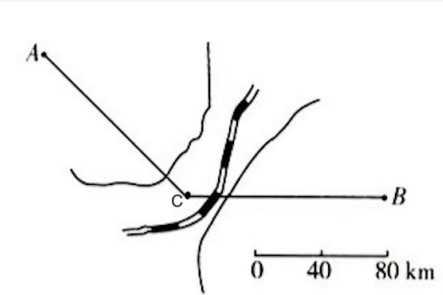

**Question:** Measure the distances of AC and CB according to the scale. Driving from point A to point C and then from point C to point B, with a speed of 60 km/h, how many hours are needed? Answer with a single number, such as 1, 2, 3.

**Ground Truth: 4.**

**Response from o3-TU:** 4

**Response from o3: 3**.

**Response from o4-mini-TU:** 3

**Response from o4-mini:** 2

**Response from Gemini-2.5-Pro:** By measuring the distances using the provided scale, we find that the line segment CB is equal in length to the scale bar representing 80 km, so Distance CB = 80 km. The segment AC is shorter than the full 80 km scale; it appears to be three-quarters of the scale, giving Distance AC = $0.75 \times 80 = 60$ km, or equivalently, 1.5 times the 0–40 km portion of the scale, which also gives 60 km. Therefore, the total distance is Total Distance = Distance AC + Distance CB = $60 + 80 = 140$ km. Assuming a driving speed of 60 km/h, the time needed is Time = $\frac{\text{Total Distance}}{\text{Speed}} = \frac{140}{60} \approx 2.33$ hours, which rounded to the nearest whole number gives 2 hours.

**Response from Gemini-2.5-flash:** To solve this problem, we first determine the map scale: the scale bar shows that 223 pixels correspond to 80 km, so 1 pixel $\approx 0.3587$ km. The pixel coordinates are $A(112, 102)$, $C(352, 417)$, and $B(686, 417)$. Using the distance formula, the pixel length of $AC$ is $\sqrt{(352 - 112)^2 + (417 - 102)^2} \approx 396.01$ pixels, and $CB$ is $\sqrt{(686 - 352)^2 + (417 - 417)^2} = 334$ pixels. Converting to kilometers, $AC = 396.01 \times \frac{80}{223} \approx 142.06$ km and $CB = 334 \times \frac{80}{223} \approx 119.82$ km. Thus, the total distance is $142.06 + 119.82 = 261.88$ km. With a driving speed of 60 km/h, the travel time is $\frac{261.88}{60} \approx 4.36$ hours, which rounds to 4 hours. Therefore, the final answer is $\boxed{4}$.

- **Referring Object Proportion VQA**: Color-Channel Histogram Drawing, crop, display image with scale guide line, Color Space Conversion, Color Thresholding, Mask Visualization, Mask Proportion Calculation, Drawing a Bounding Box,

- **Rotated Image OCR**: Image Rotation, Image Display, Enhances the contrast of an image.

- **Symbolic Reasoning**: Cropping, Zooming into Regions, Pixel color analysis, Grayscale Conversion, Edge Detection, Pixel Thresholding, Corner Detection, Pixel Counting, Contour Detection, Pixel Value Plotting, Grid Line Detection.

- **Maze**: Crop, Image display with scale guide line, pixel color counting, pixel searching, drawing points.

- **Math Problems**: Color Thresholding, Cropping, Pixel Counting, Ratio Calculation, Pixel Color Counting, Marks specific points on the image, Drawing lines, Zooming, Plotting Color Channels, Image display with scale guide line, Drawing a Rectangle, Displaying with a Scale Guide.

- **Word Search Puzzle**: Cropping, Text Extraction via OCR, pixel comparison, horizontal and vertical pixel projections, Converts the image to grayscale, Pixel Value Inspection, Two Images Difference Calculation, Pixle Peak Detection, Pixel Intensity Summation, Outlier Detection.

- **Low-Light Image VQA**: Cropping, Image Brightness and Contrast Enhancement, Image Displaying, Zooming.

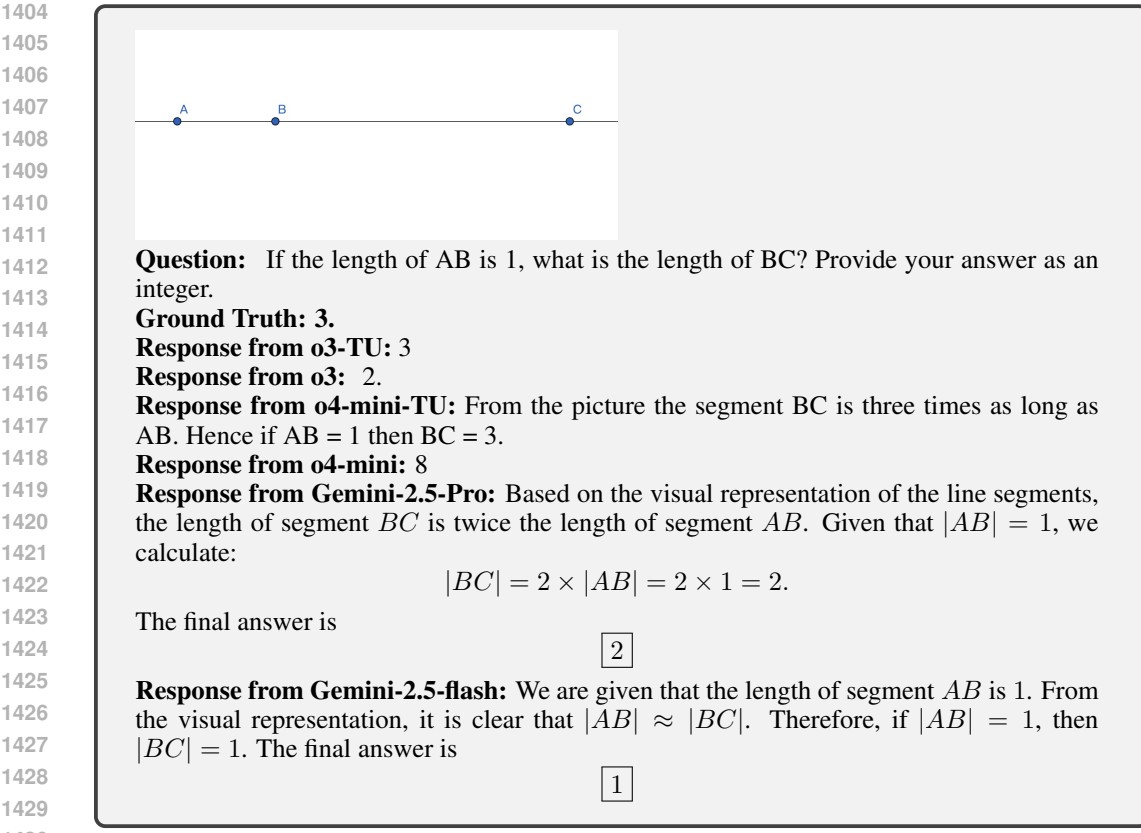

**Question:** If the length of AB is 1, what is the length of BC? Provide your answer as an integer.
**Ground Truth: 3.**
**Response from o3-TU:** 3
**Response from o3:** 2.
**Response from o4-mini-TU:** From the picture the segment BC is three times as long as AB. Hence if AB = 1 then BC = 3.
**Response from o4-mini:** 8
**Response from Gemini-2.5-Pro:** Based on the visual representation of the line segments, the length of segment $BC$ is twice the length of segment $AB$. Given that $|AB| = 1$, we calculate:
$$|BC| = 2 \times |AB| = 2 \times 1 = 2.$$
The final answer is

$$\boxed{2}$$

**Response from Gemini-2.5-flash:** We are given that the length of segment $AB$ is 1. From the visual representation, it is clear that $|AB| \approx |BC|$. Therefore, if $|AB| = 1$, then $|BC| = 1$. The final answer is

$$\boxed{1}$$

Figure 21: Examples of model responses.

- **Instrument Reading**: Cropping, Displaying, Zooming.
- **Spot the Difference**: Patch Division, Image Patch Comparison, Difference Detection, Image Splitting, Cropping, Image and Pixel Thresholding, Image Displaying.
- **Jigsaw Puzzle**: Cropping, Image Displaying, Image Patch Rearranging or Reconstruction.
- **Visual Search**: Cropping, Displaying, Zooming, Contrast Enhancement.
- **Rotation Game**: Image Rotation, Image displaying, Cropping, Drawing Reference Line, Grayscale Conversion, Pixel Thresholding, Principal Component Analysis (PCA), Angle Calculation.

## J    EFFICIENCY ANALYSIS

We further investigate the efficiency of the agentic framework by analyzing the **number of interaction turns** across tasks. The result of the number of interaction turns for PyVision models is shown in Figure 26. On average, PyVision requires **4.72** turns per problem to arrive at a solution. However, a detailed breakdown reveals an inverse correlation between interaction length and task success. For tasks where the model demonstrates mastery, the reasoning process is decisive and efficient: **Rotated OCR** and **Color VQA** require only **3.12** and **3.23** turns respectively, yet achieve state-of-the-art accuracies of 63.3% and 53.0%. In contrast, tasks with the highest interaction overhead often signal ineffective reasoning or error loops rather than successful deep thinking. The **Maze** and **Word Search** tasks incur the highest computational costs—averaging **8.58** and **7.34** turns—but yield the lowest performance (15.8% and 10.0%). This indicates that while the agentic loop allows for self-correction, prolonged interactions (exceeding ∼5 turns) often reflect the model struggling to find a valid solution path.

```
5 5 5 5 5 5 5 5 5 5 5 5 5 5 5 5 5 5 5 5
5 5 5 5 5 5 5 5 5 5 5 5 5 5 5 5 5 5 5 5
5 5 5 5 5 5 5 5 5 5 5 5 5 5 5 5 5 5 5 5
5 5 5 5 5 5 5 5 5 5 5 5 5 5 5 5 5 5 5 5
5 5 5 5 5 5 5 5 5 5 5 5 5 5 5 5 5 5 5 5
5 5 5 5 6 5 5 5 5 5 5 5 5 5 5 5 5 5 5 5
5 5 5 5 5 5 5 5 5 5 5 5 5 5 5 5 5 5 5 5
5 5 5 5 5 5 5 5 5 5 5 5 5 5 5 5 5 5 5 5
5 5 5 5 5 5 5 5 5 5 5 5 5 5 5 5 5 5 5 5
5 5 5 5 5 5 5 5 5 5 5 5 5 5 5 5 5 5 5 5
5 5 5 5 5 5 6 5 5 5 5 5 5 5 5 5 5 5 5 5
5 5 5 5 5 5 5 5 5 5 5 5 5 5 5 6 5 5 5 5
5 6 5 5 5 5 5 5 5 5 5 5 5 5 5 5 5 5 5 5
5 5 5 5 5 5 5 5 5 5 5 5 5 5 5 5 5 5 5 5
5 5 5 5 5 5 5 5 5 5 5 5 5 5 5 5 5 5 5 5
5 5 5 5 5 5 5 5 5 5 5 5 5 5 5 5 5 5 5 5
5 5 5 5 5 5 5 5 5 5 5 5 5 5 5 5 5 5 5 5
5 5 5 5 5 5 5 5 5 5 5 5 5 5 5 5 5 5 5 5
5 5 5 5 5 5 5 5 5 5 5 5 5 5 5 5 5 5 5 5
5 5 5 5 5 5 5 5 5 5 5 5 5 5 5 5 5 5 5 5
```

**Question: How many times does the number 6 appear in this image? Answer with an integer, such as 1,2,3.**
**Ground Truth: 4.**
**Response from o3-TU:** 4
**Response from o3:** 10
**Response from o4-mini-TU:** 4
**Response from o4-mini:** 2
**Response from Gemini-2.5-Pro:** 8
**Response from Gemini-2.5-flash:** Let's carefully examine the image to count the occurrences of the number 6.

Scanning the original image row by row: 1. Row 1: All 5s. 2. Row 2: All 5s. 3. Row 3: All 5s. 4. Row 4: All 5s. 5. Row 5: All 5s. 6. Row 6: 5555**6**5555555555 (Found 1 '6') 7. Row 7: All 5s. 8. Row 8: All 5s. 9. Row 9: All 5s. 10. Row 10: All 5s. 11. Row 11: 55555**6**555555555 (Found 1 '6') 12. Row 12: All 5s. 13. Row 13: All 5s. 14. Row 14: All 5s. 15. Row 15: All 5s. 16. Row 16: 5**6**55555555555 (Found 1 '6') 17. Row 17: All 5s. 18. Row 18: All 5s. 19. Row 19: All 5s. 20. Row 20: All 5s.

By carefully checking all rows, we find three instances of the number 6. The provided crops also highlight these same three instances.

The total number of times the number 6 appears in the image is 3.

Figure 22: Examples of model responses.

**Question: In the figure, in which row and column does the number 9 appear? The final answer should output two number list. The first number represents the row count from top to bottom, and the second number represents the column count from left to right. For example, [1, 2] means the first row and the second column.**

**Ground Truth: [54, 50].**

**Response from o3-TU:** [54, 50]

**Response from o3:** [27, 80]

**Response from o4-mini-TU:** The odd-one-out "9" falls in the 54th row from the top and the 50th column from the left. So the answer is

[54, 50]

**Response from o4-mini:** The grid only contains the digit "4" in every position— there is no "9" anywhere. Hence no row/column coordinate can be given for a 9.

**Response from Gemini-2.5-Pro:** 1. **Analyze the image:** The image is a large grid composed almost entirely of the number '4'. The task is to find the single occurrence of the number '9'. The hint images, particularly the last one, help to zoom in on the relevant area.
2. **Locate the '9':** By examining the last hint image, we can clearly see the number '9' embedded among the '4's.
3. **Determine the row:** We need to find the row number of the '9' in the original, full image. We count the rows from the top down, starting with row 1. * Counting the rows one by one, we find that the line containing the '9' is the 34th row from the top.
4. **Determine the column:** Now, we need to find the column number of the '9' in that 34th row. We count the numbers from left to right, starting with column 1. * In the 34th row, let's count the position of the '9': '4 4 4 4 4 4 4 4 4 4 4 4 4 4 9 ...' * Counting from the left, we can see there are 14 '4's before the '9'. Therefore, the '9' is in the 15th position.
5. **Final Answer:** The number 9 is in the 34th row and the 15th column. The output should be a list of these two numbers.

[34, 15]

**Response from Gemini-2.5-flash:** The number 9 appears in the figure. To find its location, we need to count the rows from top to bottom and the columns from left to right.
1. **Count the total number of rows:** By carefully examining the original image, we can count that there are 50 rows of numbers. 2. **Locate the row of the number 9:** Observing the last provided hint image, which clearly shows the number 9, we can see that it is located in the 8th row from the bottom of that specific crop. If we apply this to the full 50-row image, the row containing the 9 is 50 - 8 + 1 = 43rd row from the top. (Alternatively, if we count from the bottom, the 9 is in the 8th row from the bottom. So, 50 - 7 = 43rd row from the top). 3. **Locate the column of the number 9:** In the 43rd row, let's count the characters from left to right. The sequence for the 43rd row is: 4444444444444444444444444444444444494444444444 Counting from the left, the number 9 is the 38th character in this row.
Therefore, the number 9 appears in the 43rd row and the 38th column.

The final answer is $\boxed{[43, 38]}$

Figure 23: Examples of model responses.

**0. Question:**

**What is closest to the proportion of the color azure in the image? Select from the following**

**(A) 20% (B) 30% (C) 40% (D) 50%**

**1. Azure colors are defined by the threshold of the RGB channel**

*"... I'll define a condition where the blue channel is greater than 150 and the red channel is below 50, which could signify azure. ..."*

**Write code to calculate Azure percentage using defined thresholds.**

**2. Calculate Azure percentage using defined thresholds**

```
blue_mask = (arr[:,:,2] > 150) & (arr[:,:,0] < 50)
azure_pixels = np.sum(blue_mask)
total_pixels = arr.shape[0]*arr.shape[1]
azure_prop = azure_pixels / total_pixels
azure_prop
```

**4. Execution Results**
0.34511009720293595

**5. Adjust the defined thresholds of the RGB channel**

*"... thresholds might slightly vary—including edges in the analysis—so I'll consider adjusting the threshold a bit to account for minor variations. ..."*

**6. Calculate Azure percentage using adjusted thresholds**

```
blue_mask2 = (arr[:,:,2] > 120) & (arr[:,:,0] < 100)
azure_prop2 = np.sum(blue_mask2)/ total_pixels
azure_prop2
```

**7. Execution Results**
0.3684784764927594

**8. The proportion of auzre is determined by subtracting the proportion of white from the total.**

*"... To refine, I could define an "azure" pixel as one with a blue hue, excluding white (defined by RGB values >= 240 across all channels). I'll count non-white pixels and see if this helps clarify how to set the threshold ..."*

**9. Calculate Azure percentage using adjusted thresholds**

```
white_mask = (arr[:,:,0] > 240) /
           & (arr[:,:,1] > 240) /
           & (arr[:,:,2] > 240)
nonwhite_pixels = total_pixels - np.sum(white_mask)
nonwhite_prop = nonwhite_pixels / total_pixels
nonwhite_prop
```

**10. Execution Results**
0.4222376512596707

**11. Answer:**

**The azure (blue) stripes occupy a bit under half of the image—roughly four-tenths of the total pixels—so the closest option is (C) 40 %.**

Figure 24: Response from o3, case a proportion estimation from TIR-Bench.

**0. Question:**

**How many times does the number 6 appear in this image? Answer with an integer, such as 1,2,3.**

**1. Understanding the question**

*I'm thinking of using a method in Python to analyze glyphs, where I compare a grid of glyphs to a template of "5". I'll start by segmenting the grid and determining how many columns there are, assuming a 540px width and a 30px per glyph, so probably 18 columns. I'll use this template from the first cell, and then I'll compute differences between each cell and the template. If the difference is above a threshold, I'll label it as a "6". Let's test this out!*

**Few wrong step with wrong answer or meeting errors, for assuming wrong numbers of columns ....**

**2. Classify numbers by calculating differences between first glyph as template and others**

*I'm working with a 20x20 grid of glyphs, each about 12px wide, and each glyph is separated by a consistent space. I'll use the first glyph as a template to compare against the others. To find matching cells, I'll extract the average image for the first cell and check the difference using XOR calculations, since I can't simply subtract the values. If the difference exceeds a threshold, I'll mark cells as "6." Time to implement this logic and see if it works!*

**3. Check the differences**

```
diffs=[]
for cell in cells:
    h, w = cell.shape
    new=np.zeros_like(template)
    new[:h,:w]=cell
    diff = np.logical_xor(new,template).sum()
    diffs.append(diff)
min(diffs), max(diffs)
```

**4. Execution Results**
(0, 69)

**5. Check the distribution of differences**

*When the difference exceeds a threshold, a cell will be marked as "6." I'll check the distribution of differences to ensure accuracy. Time to implement!*

**6. Check the distribution of differences**

```
counts={}
for d in diffs:
    counts[d]=counts.get(d,0)+1
sorted_counts=sorted(counts.items())
sorted_counts[:10], len(diffs)
```

**7. Execution Results**
([(0, 396), (69, 4)], 400)

**8. Answer**
4

Figure 25: Response from o3, case a word searching from TIR-Bench.

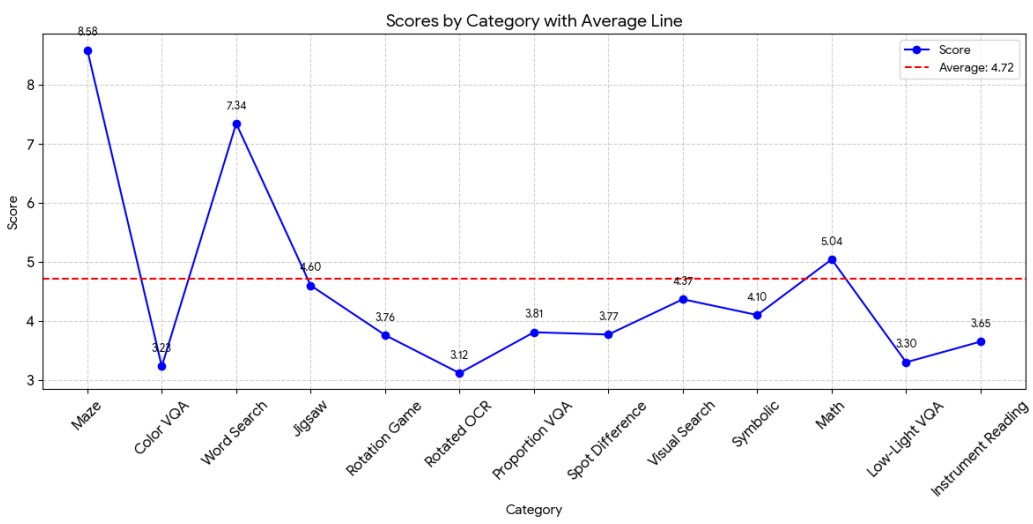

Figure 26: Number of interaction turns for the PyVision model.

