# OpenReview forum: "TIR-Bench: A Comprehensive Benchmark for Agentic Thinking-with-Images Reasoning"
_ICLR.cc/2026/Conference — Submitted to ICLR 2026_

### Official Review · Reviewer_WUfX · 2025-10-25

**Soundness:** 3
**Presentation:** 3
**Contribution:** 2
**Rating:** 4
**Confidence:** 4

**Summary:**

This paper introduces TIR-Bench, a novel and comprehensive benchmark designed to evaluate the agentic thinking-with-images capabilities of MLLMs. The authors argue that existing benchmarks focus on static image understanding and fail to assess the ability of models like O3 to actively use tools to transform and manipulate images during problem-solving. TIR-Bench comprises 13 diverse tasks that are intentionally designed to be unsolvable without tool use. The evaluation of 22 models demonstrates that the benchmark is highly challenging (46% accuracy for the best model, O3-TU) and that the availability of tool-use capabilities is the key differentiator for performance.

**Strengths:**

1. The paper identifies a significant gap in current multimodal evaluation: the lack of assessment for agentic or tool-assisted image reasoning, which is a key emergent capability of frontier models.
2. The 13 tasks in TIR-Bench are cleverly designed with a common trait: they cannot be solved by static observation alone. This strongly compels models to adopt a "think-tool-observe" loop, aligning perfectly with the benchmark's evaluation goals.
3. The experiments, especially the comparison between tool-enabled and non-tool-enabled models, clearly demonstrate that tool use is essential for these tasks and quantify the significant room for improvement, even for SOTA models.

**Weaknesses:**

1. The paper's core evaluation is on "tool use," but this is primarily equated to a "Code Interpreter" in the experiments. This makes the evaluation more of a test of a model's "Python CV coding ability" rather than a test of general-purpose "tool calling" capabilities.
2. In the experimental setup, proprietary SOTA models (O3-TU, O4-mini-TU) are granted access to a code interpreter, while open-source models are not. This is an unfair comparison, as the poor performance of open-source models might stem from the lack of a robust execution sandbox, not from inferior reasoning.
3. While the paper states many samples are newly created, a portion of the data is sourced from existing public datasets. It is difficult to guarantee that large models like O3 have not seen these or highly similar samples during pre-training.
4. The best model (O3-TU) scores only 46%, meaning it fails most of the time. The paper mentions some failures but lacks a systematic analysis of why. On which tasks do models fail most? What are the common failure modes?
5. Some tasks (e.g., Word Search, Low-Light VQA) feel overly synthetic, designed specifically to force the use of a particular tool (e.g., pixel comparison, image enhancement). While this serves the benchmark's purpose, it is debatable whether these tasks represent the diverse, unpredictable tool-use scenarios MLLMs will face in the real world.

**Questions:**

1. Many tasks in TIR-Bench seem better suited for traditional Computer Vision algorithms than for MLLMs. While models can solve them by writing code to call CV libraries, it's debatable whether this tests the MLLM's "reasoning" or its ability to recall and execute CV library APIs.
2. For a benchmark claiming to evaluate a "thinking process," the paper relies solely on final-answer accuracy. This obscures crucial details: did the model solve the problem efficiently, or via extensive, ineffective trial-and-error? Metrics for the quality, efficiency, or plausibility of the reasoning path are missing.
3. The pilot study in Sec 4.5 is conducted on only one task. The finding that "Tool-Use SFT" outperforms "Direct SFT" is insightful, but this conclusion can hardly be generalized to the other 12 complex tasks in TIR-Bench based on this single experiment.
4. See weakness.

---

> ### Author Response · Authors · 2025-11-21
> **Response Part 1**
>
> We sincerely thank the reviewer for the careful and thoughtful comments and the time on reviewing our paper.  We are encouraged that the reviewer recognizes that our work identifies a significant gap in current multimodal evaluation regarding agentic or tool-assisted image reasoning. Below are responses to all the questions as follows (W: Weakness; Q: Question; A: Answer):
>
> >W1: The paper's core evaluation is on "tool use," but this is primarily equated to a "Code Interpreter" in the experiments. This makes the evaluation more of a test of a model's "Python CV coding ability" rather than a test of general-purpose "tool calling" capabilities.
>
> **R1: (1). Scope – Thinking with Images:** We respectfully clarify that the goal of this benchmark is to evaluate the model’s ability to think with images, which refers to **Visual Tool Use** performing diverse image operations such as cropping, zooming, editing, and spatial manipulation and incorporate information of processed images into reasoning [1,2,3], rather than general tool usage such as web searching.
>
>  **(2). Code as a Universal Tool:** In current think-with-images area, there are two main strategies of processing images: (a) generating code to create and use tools such as image cropping (e.g., o3, o4-mini, PyVision [1]), and (b) calling predefined image manipulation tools (e.g., Deepeyes [2] and mini-o3 [4]). All the tools relevant to this paper and benchmark are **image-manipulation focused**. In our experiments, we evaluate both two kinds of models, such as o3-TU and Deepeyes.
>
> **(3). Why Code Interpreter Matters:** We would like to clarify that **generating code to create tools is a form of advanced tool-calling capability (i.e., calling code interpreter or runtime environment)**. It is more complex, flexible, and powerful than relying on statically pre-defined tools [1, 3], which requires the model to write code to create novel capabilities (such as crop, rotation, or drawing auxiliary lines).
>
> > W2: In the experimental setup, proprietary SOTA models (O3-TU, O4-mini-TU) are granted access to a code interpreter, while open-source models are not. This is an unfair comparison, as the poor performance of open-source models might stem from the lack of a robust execution sandbox, not from inferior reasoning.
>
> **R2:** We appreciate this suggestion and wish to clarify that open-source models (e.g., LLaVA) were not denied access to an interpreter; they were evaluated in their native, static-observation state because they are not fine-tuned for agentic code interpreter use. Equipping an open-source MLLM with a code interpreter or tool calling ability is a highly non-trivial technical challenge.
>
> **(1) Training Limitations:** Unlike proprietary models (e.g., O3-TU) optimized for agentic tool use [1], most open-source MLLMs (e.g., LLaVA, InternVL) lack the fine-tuning required to utilize external sandboxes effectively. While enabling "think-with-images" capabilities is an active research direction (e.g., Deepeyes [2]), robust implementation remains difficult and challenging. Therefore, our primary evaluation focused on the models' native, as-released capabilities.
>
> **(2) New Experiments:** To address the reviewer's request regarding larger models, we adopted a runtime strategy and well-designed prompt similar to PyVision [3]. We re-implemented PyVision for **Qwen2.5-VL-72B** and **Qwen3-VL-235B** by providing them with a code-execution environment. As shown in updated **Table 2**, their scores improved to **24.0** and **24.9**, respectively. These results indicate that adding a runtime environment can indeed boost performance for large open MLLMs, though we observed that approximately **10%** of the generated code remained non-executable. For small open-source models such as Qwen2.5-VL-7B, most of the generated code is non-executable in our experiments.

---

> > ### Author Response · Authors · 2025-11-21
> > **Response Part 2**
> >
> > > W3: While the paper states many samples are newly created, a portion of the data is sourced from existing public datasets. It is difficult to guarantee that large models like O3 have not seen these or highly similar samples during pre-training.
> >
> > **R3:** This is a critical concern for all modern benchmarks. We have mitigated this in two ways: **(1)** The majority of our dataset is newly created (e.g., math, symbolic problems). For any tasks adapted from public datasets, most problems were fundamentally altered and the questions are newly-created to require image manipulation (e.g., referCOCO and OCRbench). Any memorized answer from the original static source is thus useless for solving our adapted version. As a result, more than **90%** of the new problems in TIR-Bench are newly created ensuring our benchmark measures "genuine generalization" for existing models. **(2)** For the remaining data taken directly from ColorBench, we note that the release of ColorBench was close to the release of o3, so the probability of data contamination in ColorBench is very low. **In summary, we believe that TIR-Bench has effectively mitigated contamination issues.**
> >
> >
> > >W4: The best model (O3-TU) scores only 46%, meaning it fails most of the time. The paper mentions some failures but lacks a systematic analysis of why. On which tasks do models fail most? What are the common failure modes?
> >
> > **R4:** Thank you for the suggestion. Overall, o3-TU performs worst on the **Jigsaw Puzzle** and **Instrument Reading** tasks. However, we note that the OpenAI API for o3 and o4-mini does not return the models’ internal reasoning traces, which prevents us from systematically analyzing their failure modes. The examples shown in the paper were obtained by manually running the models through the web interface. However, we cannot run all data through the web interface. Therefore, we use PyVision, a GPT-4.1–based mimic of o3-TU for analysis. We found that approximately **37%** of PyVision’s erroneous responses are due to **perception errors**, such as **cropping the wrong image patch** or **perceive images wrong**, while **61%** are **tool generation and reasoning errors**, including **language reasoning mistakes** or **errors in reasoning about writing code for image processing,** e.g., the generated tools can not solve the problems. The remaining 2% are other errors.
> >
> > >W5: Some tasks (e.g., Word Search, Low-Light VQA) feel overly synthetic, designed specifically to force the use of a particular tool (e.g., pixel comparison, image enhancement). While this serves the benchmark's purpose, it is debatable whether these tasks represent the diverse, unpredictable tool-use scenarios MLLMs will face in the real world.
> >
> > **R5:** We acknowledge this limitation but it is a deliberate methodological choice. As the first benchmark in the Think-With-Images area, we aim to cover as **many image-manipulation abilities** as possible. To robustly and reliably evaluate a specific agentic image-manipulation capability (e.g., pixel-level search, image enhancement), it is necessary to design tasks that isolate that capability.
> >
> > **(1) Low-Light VQA** is grounded in real-world scenarios, as many images are captured under poor lighting and require enhancement; the task uses images from established real-world enhancement datasets.
> >
> > **(2) Word search** is a common puzzle format. About 20% of our tasks come from the Internet, while the rest are synthesized to expand the dataset, following practices used in benchmarks like LogicVista and PuzzleVQA.
> >
> > **(3)** Many tasks in TIR-Bench such as instrument reading, visual search, math, and rotated OCR are grounded in **real-world scenarios**, making the benchmark well-suited for evaluating the diverse real-world tool-use abilities of MLLMs.
> >
> > **We have added this to the limitations section (Section 6) and leave further real-world examples for certain tasks for future work.**

---

> > > ### Author Response · Authors · 2025-11-21
> > > **Response Part 3**
> > >
> > > >Q1: Many tasks in TIR-Bench seem better suited for traditional Computer Vision algorithms than for MLLMs. While models can solve them by writing code to call CV libraries, it's debatable whether this tests the MLLM's "reasoning" or its ability to recall and execute CV library APIs.
> > >
> > > **A1:** (1) We respectfully clarify that **think-with-images** reasoning **differs** from traditional language-based reasoning. This paradigm involves calling pre-defined tools or writing code to create tools to manipulate images and subsequently integrating the processed image information into the reasoning process [1, 2, 3]. Consequently, the ability to call, write, and execute code or pre-defined tools for image manipulation is a fundamental component of this specific visual reasoning capability compared with language reasoning.
> > >
> > > (2) We note that most problems in TIR-Bench are VQA questions that cannot be solved by traditional Computer Vision (CV) algorithms alone (e.g., tasks involving math, symbolic problems, or mazes). Although some tasks, such as low-light VQA, necessitate calling traditional CV algorithms or writing new code to implement them, this is by design. This approach intentionally tests the "think-with-images" capacity to create image-manipulation tools (via code) or utilize pre-defined tools (e.g., image enhancement or cropping tools). In summary, implementing or calling pre-defined CV algorithms (like crop) is indeed part of "think-with-images" reasoning.
> > >
> > > > Q2: For a benchmark claiming to evaluate a "thinking process," the paper relies solely on final-answer accuracy. This obscures crucial details: did the model solve the problem efficiently, or via extensive, ineffective trial-and-error? Metrics for the quality, efficiency, or plausibility of the reasoning path are missing.
> > >
> > > **A2: (1). Clarification:** We would like to respectfully clarify that our paper and benchmark do not aim to evaluate the quality of the thinking process of MLLMs. Instead, they focus on assessing the think-with-images capability of agentic MLLMs. The general thinking processes of MLLMs and thinking-with-images reasoning are fundamentally different. think-with-images capability of agentic MLLMs refer to writing code to create tools to manipulate images (e.g., PyVision [1] and o3) or calling pre-defined tools (e.g., calling crop tool or zoom in tool in Deepeyes [2]).
> > >
> > > **(2).More Analysis:**  Following the reviewer's comments, we evaluated PyVision's efficiency by analyzing interaction turns. The results and analysis are detailed in Appendix J and Figure 26 in the updated paper. We briefly discuss them here. On average, the model requires **4.72** turns per problem. **We observe that efficiency correlates with proficiency**: Rotated OCR and Color VQA achieve SOTA accuracy (63.3% and 53.0%) with minimal turns (3.12 and 3.23). Conversely, high interaction counts often reflect error loops rather than deep reasoning. The Maze and Word Search tasks require the most turns (8.58 and 7.34) but yield the lowest accuracy (15.8% and 10.0%). This suggests that **while the agentic loop allows for self-correction, interactions exceeding ∼5 turns often indicate an inability to find a valid solution path.**

---

> ### Author Response · Authors · 2025-11-21
> **Response Part 4**
>
> > Q3: The pilot study in Sec 4.5 is conducted on only one task. The finding that "Tool-Use SFT" outperforms "Direct SFT" is insightful, but this conclusion can hardly be generalized to the other 12 complex tasks in TIR-Bench based on this single experiment.
>
> **A3: (1)**. Following the reviewer’s suggestion, we conducted additional experiments on InternVL3-8B and evaluated its performance on the rotation game task. The results are presented in the table below. We observe the same overall trend: **agentic SFT consistently outperforms traditional direct SFT.**
>
> | Task                          | Model              | OCR    | Rotation Game |
> |------------------------------|--------------------|--------|----------------|
> | Direct SFT                   | Qwen-VL-2.5-7B     | 44.59  | 20.00         |
> | Agentic SFT                  | Qwen-VL-2.5-7B     | 84.47  | 45.33         |
> | Direct SFT                   | InternVL3-8B       | 13.33  | 34.67         |
> | Agentic SFT                  | InternVL3-8B       | 78.33  | 50.67         |
>
> **(2).** We acknowledge that the experiment may not be generalizable. However, We respectfully clarify that this experiment serves as a **pilot** study to compare agentic and vanilla SFT. A comprehensive exploration requires addressing complex challenges in data synthesis and agentic training, **which are distinct research topics beyond the scope of a benchmark paper**. Our goal was simply to discuss future research direction and demonstrate TIR-Bench's ability to quantify the performance delta between these approaches. We have added a **Limitations** section to explicitly scope the generalization of these preliminary results.
>
> **References**
>
> [1] PyVision: Agentic Vision with Dynamic Tooling.
>
> [2] DeepEyes: Incentivizing "Thinking with Images" via Reinforcement Learning.
>
> [3] Thinking with images. OpenAI.
>
> [4] Mini-o3: Scaling Up Reasoning Patterns and Interaction Turns for Visual Search.

---

> > ### Comment · Reviewer_WUfX · 2025-11-27
> >
> > I sincerely appreciate the authors for their response. It has addressed my concerns. I will update my rating to 6.

---

> > > ### Author Response · Authors · 2025-11-27
> > >
> > > We sincerely thank you once again for your insightful comments and suggestions, and for kindly increasing your score.

---

### Official Review · Reviewer_Gpt1 · 2025-10-31

**Soundness:** 2
**Presentation:** 3
**Contribution:** 3
**Rating:** 4
**Confidence:** 4

**Summary:**

The authors present TIR-Bench, a benchmark for code/tool-assisted visual reasoning on images. They identify that previous benchmarks only require simple tools at most. They curate the benchmark for tasks that require complex code/tool-assisted reasoning, with a focus on dynamic or agentic approaches, with a combination of new problems and those from existing datasets. The results suggest all models have difficulty with these tasks.

**Strengths:**

- The paper addresses a critical need for new benchmarks in this area.
- It identifies more difficult tasks than previous datasets suitable for new code-execution based approaches.
- It highlights issues with current state-of-the-art models.
- Table 2 presents a wide selection of model evaluations.
- Many qualitative examples from the benchmark are shown.
- Preliminary experiments on SFT are presented.

**Weaknesses:**

- The data seems to need more verification; annotation by one student without at least one more pair of eyes checking may not be reliable.
- Synthetic and hand-annotated data are mixed here, when they seem to evaluate markedly different capabilities. Perhaps they should be switched?
- The provenance of the data is unclear, posing potential issues for usage.
- The figures showing model responses with interleaved thinking/code, like 4, 5, 24, 25 are difficult to parse.

**Questions:**

- The function calling and code comparison is interesting, but is the primary difference actually the presence of predefined tools rather than whether function calling or full code generation is used? This should be clarified more.

---

> ### Author Response · Authors · 2025-11-21
> **Response Part 1**
>
> Thank you for the reviewer's time and effort in reviewing our paper and constructive comments.  We are encouraged that the reviewer recognizes that our work addresses a critical need for new benchmarks in the area of code/tool-assisted visual reasoning. Below are the responses to the reviewer's questions or comments.
>
> > W1: The data seems to need more verification; annotation by one student without at least one more pair of eyes checking may not be reliable.
>
> **R1:** We acknowledge and appreciate your concern regarding data integrity. To clarify our stringent verification methods: For all **newly-annotated** problems, we employed **two PhD students** to perform the initial annotation. Subsequently, both students engaged in a **reciprocal cross-checking** process to verify the correctness of both the problems and their respective answers (as detailed on **lines 208 and 242**). For problems drawn from **other established benchmarks** (e.g., ColorBench), while a single student was tasked with the selection process, these problems inherit reliability from their prior validation in existing academic work. Following the reviewer’s suggestion, we commissioned **two additional students** to **re-check** the benchmark problems during the rebuttal, and no errors were found. Based on this dual verification strategy, we assert the **reliability** of the problems included in this benchmark. **The paper has been revised and updated to offer a clearer description of the annotation and checking procedures.**
>
> > W2: Synthetic and hand-annotated data are mixed here, when they seem to evaluate markedly different capabilities. Perhaps they should be switched?
>
> **R2:** We appreciate this insightful observation and wish to clarify that the division between synthetic and hand-annotated data is task-based. For instance, Math, Visual Search, and Instrument Reading are **fully hand-annotated**, whereas Maze and Word Search are **fully synthetic**. Users can therefore easily compare model performance by analyzing the results per task.
>
> We have also adopted the reviewer’s suggestion and updated our presentation to include **explicit sub-benchmarks**: **TIR-Bench-Hand-Annotated** and **TIR-Bench-Synthetic**, for a direct comparison. Please refer to **Appendix F** for the experimental results and additional analysis.
>
> > W3: The provenance of the data is unclear, posing potential issues for usage.
>
> **R3:** We appreciate this insightful suggestion. We have revised the paper to provide a detailed **Data Provenance Statement** in the **Appendix B** and **Section 3** for all 13 tasks. Data from existing benchmarks are covered by the **Apache License 2.0** or other licenses that **permit academic use**. For newly annotated data, images were sourced from freely available platforms like **Wikimedia Commons **and the **National Gallery of Art**, or were created by students using tools such as **GeoGebra**. Consequently, the usage of this benchmark is **fully permissible**.
>
> **We have revised the paper and added detailed sources for each task in Appendix B and Section 3.**

---

> > ### Author Response · Authors · 2025-11-21
> > **Response Part 2**
> >
> > > W4: The figures showing model responses with interleaved thinking/code, like 4, 5, 24, 25 are difficult to parse.
> >
> > **R4:** We thank the reviewer for this feedback. We have redesigned these figures in the updated revision, using clearer formatting (e.g., colored blocks, distinct typography) to visually separate the natural language reasoning, code execution, and image observation steps, making the agent's trajectory much easier to follow.
> >
> > >Q1: The function calling and code comparison is interesting, but is the primary difference actually the presence of predefined tools rather than whether function calling or full code generation is used? This should be clarified more.
> >
> > **A1:** The observation is correct. This experiment aims to evaluate two main strategies used in the current think-with-images area: **(1)** generating code to create and use tools (e.g., o3, o4-mini, PyVision [1]), and **(2)** calling predefined tools (e.g., Deepeyes [2]).
> >
> > We assert that the difference between these two approaches is **fundamental**, resting on the necessity for models to either **create tools themselves** or rely on the presence of **predefined tools**. Function Calling represents a simpler, typically stateless mechanism for invoking a fixed set of predefined APIs. However, Code Generation for Tool Generation, which requires the model to write code to create novel capabilities (such as crop, rotation, or drawing auxiliary lines), is far more complex and powerful. This latter ability serves to test an agent's capacity for dynamically writing and executing novel tools in a stateful environment, following the design principles of o3 and PyVision [1]. Given that function calling is less powerful but more readily learned by models, the experiments in this part are crucial for checking the differential abilities of MLLMs in both utilizing static tools through function calling and generating dynamic tools through code writing.
> >
> > **References:**
> >
> > [1] PyVision: Agentic Vision with Dynamic Tooling.
> >
> > [2] DeepEyes: Incentivizing "Thinking with Images" via Reinforcement Learning.

---

### Official Review · Reviewer_ALVA · 2025-10-31

**Soundness:** 2
**Presentation:** 3
**Contribution:** 2
**Rating:** 4
**Confidence:** 4

**Summary:**

The paper presents TIR-Bench, a new benchmark designed to evaluate the agentic visual reasoning ability of multimodal large language models (MLLMs). Such a thinking-with-images process requires the models to actively manipulate images as part of the reasoning process rather than static analysis. The benchmark consists of 13 tasks, such as rotation, maze solving, and jigsaw puzzles, and the dataset has 1,215 examples. The authors evaluate 22 MLLMs, including open-source, proprietary, and tool-using models. Results show that the TIR-Bench is universally challenging, with 46% accuracy from o3-TU as the performance. It reveals that non-agentic models perform substantially poorly, highlighting the necessity of tool-based reasoning for complex visual tasks. Finally, the authors also study function calling behaviors and run a pilot fine-tuning comparison on rotated-OCR, finding that agentic/tool-use SFT outperforms direct SFT.

**Strengths:**

- The task design is comprehensive, covering 13 diverse tasks that span a broad range of visual reasoning abilities. Each task forces models to manipulate images rather than passively describing them, enabling a faithful evaluation of thinking-with-images reasoning.
- The benchmark is sufficiently challenging and ensures it is a useful, long-term suite for measuring MLLMs’ reasoning ability.
- A central contribution of TIR-Bench is that it explicitly investigates how models interact with images, which is crucial to understand the model’s agent use behavior and assess their ability for visual reasoning tasks.

**Weaknesses:**

- TIR-Bench is designed to evaluate models’ tool-based visual reasoning ability, but many evaluated models (e.g., LLaVA, Qwen2.5-VL, InternVL) cannot execute code or invoke tools. Their inclusion mainly serves as a static baseline rather than a true test of “thinking-with-images.”
- 1215 examples are divided into 13 tasks, making several tasks have modest data sizes. It raises concerns about the benchmark robustness given the data scale.
- Using GPT-4o to extract final answers can introduce parsing bias. However, the authors do not verify GPT-4o’s extraction accuracy. The influence of the extraction process should be excluded.
- The agentic vs direct SFT pilot uses Qwen-2.5-VL-7B only for a single task. Broader models and tasks are needed to generalize the conclusion.
- Some models can use external tools during reasoning, while others do not have access. This discrepancy makes it difficult to interpret the model’s intrinsic reasoning capability and limitations. The authors should clarify what external tools are accessible for each model.
- While the paper defines thinking with images as a process in which models manipulate images through tool use, the results in Section 4.4 indicate that models do not always invoke tools unless explicitly guided by the prompt. This raises questions about whether the poorly performing models fail due to insufficient reasoning ability or simply because they were not prompted to use available tools.

**Questions:**

- Could the authors clarify the rationale for including non-agentic models for tool-using ability, given that they can not use tools or write code and execute?
- Could the authors justify whether the data scale is sufficient to get statistically meaningful and reliable conclusions?
- Could the authors validate the effectiveness of using GPT-4o for answer extraction?
- Could the authors provide the fine-tuning comparison on more models and different tasks?
- For tasks that inherently benefit from external tools, please report the tools the models use.
- Could the authors provide the exact prompts used during benchmarking and clarify to what extent the models are instructed or encouraged to use tools?

---

> ### Author Response · Authors · 2025-11-21
> **Response Part 1**
>
> We would like to express our sincere gratitude for the reviewer's efforts and time in reviewing our submission. We are encouraged that the reviewer acknowledges TIR-Bench as a sufficiently challenging and useful long-term suite for measuring the reasoning ability of MLLMs. Below are the responses to the reviewer's questions or comments.
>
> > W1: TIR-Bench is designed to evaluate models’ tool-based visual reasoning ability, but many evaluated models (e.g., LLaVA, Qwen2.5-VL, InternVL) cannot execute code or invoke tools. Their inclusion mainly serves as a static baseline rather than a true test of “thinking-with-images.”
>
> **R1:** We would like to respectfully clarify that the inclusion of traditional MLLMs such as LLaVA is not intended to evaluate their thinking-with-images capabilities, but rather to demonstrate that without such capabilities, MLLMs perform poorly on TIR-Bench. As detailed in Section 4.2, these non-agentic baselines are essential for underscoring the significance of agentic reasoning. Indeed, our results show that traditional models perform near chance levels on TIR-Bench—for example, the best-performing non-agentic model (Gemini-2.5-Pro) achieves only ~29% accuracy, barely surpassing random guessing. This stark underperformance reinforces that TIR-Bench effectively targets genuine thinking-with-images ability. Therefore, the inclusion of non-agentic models is not only justified but fundamental to validating the benchmark’s purpose and claims.
>
> > 1215 examples are divided into 13 tasks, making several tasks have modest data sizes. It raises concerns about the benchmark robustness given the data scale.
>
> **R2:** We appreciate this valid point regarding data scale. We offer four key points of clarification:
>
> **1. Complexity over Scale:** Our 1,215 samples represent **complex, multi-step agentic** trajectories, not simple QA pairs. The entire benchmark pipeline from data curation and verification to the model evaluation itself is **extremely resource-intensive**. This scale **is consistent with or exceeds** other seminal trajectory benchmarks (e.g., OSWorld [1]).
>
> **2. Empirical Stability:** To directly address concerns about statistical robustness, we conducted repeated evaluations using the o4-mini model. Across multiple runs, the performance variance was **less than 3%**. This low variance confirms that the benchmark yields highly **stable and reproducible** signals, rather than random fluctuations.
>
> **3. Discriminative Power:** While the tasks are difficult, the benchmark is statistically robust in its ability to differentiate models. We observe **consistent, significant performance gaps** between model families (e.g., agentic vs. non-agentic), which confirms that the sample size is sufficient to measure clear capability differences without being dominated by noise.
>
> **4. Holistic Evaluation:** While the 13 sub-tasks vary in size, the benchmark is designed to be evaluated as a whole. The 1,215-sample dataset collectively provides a strong signal for the challenging 'think with image' capabilities we aim to measure.
>
> > W3: The authors do not verify GPT-4o’s extraction accuracy. The influence of the extraction process should be excluded.
>
> **R3:** We appreciate this important point. Our use of GPT-4o aligns with the validated "LLM-as-a-Judge" paradigm [2], which is a state-of-the-art practice for scalable benchmark evaluation. Following the reviewer’s suggestion, we hired several students to verify the results of o3 and Gemini-2.5-Pro, and the consistency of extracted answers exceeds **96%**.

---

> > ### Author Response · Authors · 2025-11-21
> > **Response Part 2**
> >
> > > W4: The agentic vs direct SFT pilot uses Qwen-2.5-VL-7B only for a single task. Broader models and tasks are needed to generalize the conclusion.
> >
> > **R4: (1).** Following the reviewer’s suggestion, we conducted additional experiments on InternVL3-8B and evaluated its performance on the rotation game and OCR tasks. The results are presented in the table below. We observe the same overall trend: agentic SFT **consistently outperforms traditional direct SFT**.
> >
> > | Task        | Model          | OCR   | Rotation Game |
> > | ----------- | -------------- | ----- | ------------- |
> > | Direct SFT  | Qwen-VL-2.5-7B | 44.59 | 20.00         |
> > | Agentic SFT | Qwen-VL-2.5-7B | 84.47 | 45.33         |
> > | Direct SFT  | InternVL3-8B   | 13.33 | 34.67         |
> > | Agentic SFT | InternVL3-8B   | 78.33 | 50.67         |
> >
> > **(2).** We respectfully clarify that this experiment serves as a **pilot study** to compare agentic and vanilla SFT. A comprehensive exploration requires addressing complex challenges in data synthesis and agentic training, **which are distinct research topics beyond the scope of a benchmark paper**. Our goal was simply to discuss future research direction and demonstrate TIR-Bench's ability to quantify the performance delta between these approaches. We have added a **Limitations section** to explicitly scope the generalization of these preliminary results.
> >
> > > W5: Some models can use external tools during reasoning, while others do not have access. This discrepancy makes it difficult to interpret the model’s intrinsic reasoning capability and limitations. The authors should clarify what external tools are accessible for each model.
> >
> > **R5:** We adhered strictly to the original tool definitions for each model. Specifically, **o3-TU**, **o4-mini-TU**, and **PyVision** were evaluated in an identical setup: **a sandboxed Code Interpreter, where the MLLM itself can create any tools by coding**, without pre-define any tool primitives . **Deepeyes** was evaluated using its **own native toolset** (e.g., **cropping** and **zooming**). All other traditional MLLMs were assessed using **language reasoning** but **without** agentic tool reasoning. We would like to emphasize that training open-source MLLMs with think-with-images ability (either with code interpreter like o3 or pre-defined tools such as cropping like Deepeyes) is challenging and is a main research direction in this area [3,4]. Therefore, naively prompting MLLMs with a sandboxed Code Interpreter or tools is meaningless. Therefore, we directly test traditional MLLMs and show that agentic think-with-images is important for TIR-Bench.
> >
> > > W6: While the paper defines thinking with images as a process in which models manipulate images through tool use, the results in Section 4.4 indicate that models do not always invoke tools unless explicitly guided by the prompt. This raises questions about whether the poorly performing models fail due to insufficient reasoning ability or simply because they were not prompted to use available tools.
> >
> > **R6:** This is an insightful point, and we thank the reviewer for highlighting it. **(1).** Since OpenAI does not provide access to the internal reasoning process of o3 and o4-mini (the examples in the paper are taken from the web interface), we use **PyVision** as the model for our study. We find that, across all data, PyVision generates code for image analysis. Therefore, the poor performance is not due to a failure to prompt the use of available tools. **(2).** We also analyzed the error types of the PyVision model. We found that approximately **37%** of PyVision’s erroneous responses are due to perception errors, such as cropping the wrong image patch or perceive images wrong, while **61%** are tool generation and reasoning errors, including language reasoning mistakes or errors in reasoning about writing code for image processing, e.g., the generated tools can not solve the problems. The remaining 2% are other errors.
> >
> > >  Q1: Could the authors clarify the rationale for including non-agentic models for tool-using ability, given that they can not use tools or write code and execute?
> >
> > A1: Please refer to R1 for W1.
> >
> > > Q2: Could the authors justify whether the data scale is sufficient to get statistically meaningful and reliable conclusions?
> >
> > A2: Please refer to R2  for W2.
> >
> > > Q3: Could the authors validate the effectiveness of using GPT-4o for answer extraction?
> >
> > A3: Please refer to R3  for W3.
> >
> > >  Q4: Could the authors provide the fine-tuning comparison on more models and different tasks?
> >
> > A4: Please refer to R4  for W4.

---

> ### Author Response · Authors · 2025-11-21
> **Response Part 3**
>
> > Q5: For tasks that inherently benefit from external tools, please report the tools the models use.
>
> **A5:** Please note that agentic MLLMs fall into two categories: those that model generate code itself to create tools such as cropping, masking,saturation adjustment etc. (e.g., o3, o4-mini, PyVision) and those that use predefined tools (e.g., Deepeyes, with only cropping and zooming).
>
> **(1).** Deepeyes relies on zoom-in and crop tools. As a result, it performs well on tasks that require these operations, such as visual search, but significantly worse on other tasks. This highlights the need for MLLMs equipped with a more diverse set of predefined image tools for future work.
>
> **(2).** For o3, o4-mini, PyVision, they write code to create tools.Because the OpenAI API does not expose the internal reasoning process or generated code (unlike the examples shown in their paper, which come from the web interface), and we cannot analyze the tools used by these models, we focus our study on PyVision. We use GPT-4o to extract and summarize the created tools. We list several tasks below, and please refer to **Appendix H** for the summarized for all tasks.
>
> **1. Color VQA task** : pixel color counting, pixel searching, color thresholding, crop, drawing color channel histogram, image enhancement, converts an image to grayscale, color proportion calculation, pixel color analysis.
>
> **2. Rotated Image OCR**:  Image Rotation, Image Display, Enhances the contrast of an image.
>
> **3. Symbolic Reasoning**:  Cropping, Zooming into Regions, Pixel color analysis, Grayscale Conversion, Edge Detection, Pixel Thresholding, Corner Detection, Pixel Counting, Contour Detection, Pixel Value Plotting, Grid Line Detection.
>
> **4. Instrument Reading**: Cropping, Displaying, Zooming.
>
> **5. Spot the Difference**: Patch Division, Image Patch Comparison, Difference Detection, Image Splitting, Cropping, Image and Pixel Thresholding, Image Displaying
>
> > Q6: Could the authors provide the exact prompts used during benchmarking and clarify to what extent the models are instructed or encouraged to use tools?
>
> **A6:** We respectfully point out that the main benchmarking and function calling experiments are fundamentally different, having distinct purposes and setups.
>
> For the main benchmark, we used a **zero-shot approach that questions were fed directly without explicit instruction** to simulate real-world constraints, relying only on the **models' internal tools **(e.g., pre-defined tools or Code Interpreter). Please note that training models to recognize when and how to invoke the correct tools is an important research topic in the think-with-images area.
>
> The function calling experiments, however, were designed specifically to test the models' ability to correctly **invoke equipped, existing tools** when explicitly prompted (as detailed in **Appendix D**). We evaluate three prompts with varying degrees of guidance to assess the model’s function-calling ability (the prompts are provided in **Appendix D**). This experiment confirms the reviewer's observation that current MLLMs generally require explicit prompting to effectively utilize their tools.
>
> **References**
>
> [1] OSWorld: Benchmarking Multimodal Agents for Open-Ended Tasks in Real Computer Environments. Neurips 2024.
>
> [2] MathVista: Evaluating Mathematical Reasoning of Foundation Models in Visual Contexts. ICLR 2024.
>
> [3] DeepEyes: Incentivizing "Thinking with Images" via Reinforcement Learning.
>
> [4] Mini-o3: Scaling Up Reasoning Patterns and Interaction Turns for Visual Search.

---

### Official Review · Reviewer_8Z6B · 2025-11-01

**Soundness:** 3
**Presentation:** 3
**Contribution:** 2
**Rating:** 6
**Confidence:** 4

**Summary:**

This paper aims to better evaluate the ability to think with images in the chain of thought. Current work focuses primarily on localization and cropping capabilities. To promote and inspire broader capabilities, this paper proposes a more challenging benchmark involving thinking with images. The proposed benchmark includes 13 diverse tasks, encompassing image understanding tasks such as color setting, image selection, and maze solving. The authors evaluated open-source and proprietary models in experiments, and further evaluated the proprietary model, which can utilize image processing tools.

**Strengths:**

1. The paper elevates thinking-with-images from simple localization and cropping capabilities to more complex and diverse ones.

2. The authors set up a tool-using scenario, experimenting with state-of-the-art models o4-mini and o3 on the proposed benchmark, achieving the expected performance improvements.

**Weaknesses:**

1. Table 2 only evaluates a limited number of open-source models, covering only Illava, Qwen2.5-VL, and InternVL3. Many large-scale open-source multimodal models have not been extensively evaluated.

2. The authors did not attempt to configure tool-using on open-source models with larger parameters, such as those with over 32 bytes of parameters. Performing such evaluations on the proposed benchmark would provide a clearer understanding of the actual capabilities of the open-source models.

**Questions:**

1. Refer to the issues raised in the weakness section.
2. Figure 3 is unclear and difficult to interpret. The authors should provide a more detailed explanation.

---

> ### Author Response · Authors · 2025-11-21
> **Response to Review**
>
> We sincerely thank the reviewer for the careful, thoughtful comments and the time devoted to reviewing our paper. We are encouraged that the reviewer recognizes our work’s contribution in elevating thinking-with-images from simple localization to more complex and diverse capabilities. We provide responses to all the questions as follows (W: Weakness; Q: Question; A: Answer):
>
> > W1: Many large-scale open-source multimodal models have not been extensively evaluated.
>
> **R1:** We agree with the reviewer that a broader evaluation is beneficial. The models included were selected as they represented the state-of-the-art and most widely used open-source MLLMs at the time of our initial submission. In the revised version, we include six additional models: DeepSeek-VL-2-Think, DeepSeek-VL-2-Direct, Keye-VL-8B, Phi-4-Multimodal-Instruct, Qwen3-VL-8B-Instruct, and Qwen3-VL-8B-Thinking. Please refer to Table 2 for the corresponding results.
>
> > W2: The authors did not attempt to configure tool-using on open-source models with larger parameters, such as those with over 32 bytes of parameters. Performing such evaluations on the proposed benchmark would provide a clearer understanding of the actual capabilities of the open-source models.
>
> **R2:** We appreciate this suggestion, as it highlights a significant area of current research. However, equipping standard open-source MLLMs with a code interpreter is a non-trivial engineering challenge.
>
> **(1) Training Limitations:** Unlike proprietary models (e.g., O3-TU) optimized for agentic tool use [1], most open-source MLLMs (e.g., LLaVA, InternVL) lack the fine-tuning required to utilize external sandboxes effectively. While enabling "think-with-images" capabilities is an active research direction (e.g., Deepeyes [2]), robust implementation remains difficult and challenging. Therefore, our primary evaluation focused on the models' native, as-released capabilities.
>
> **(2) New Experiments:** To address the reviewer's request regarding larger models, we adopted a runtime strategy similar to PyVision [3]. We re-implemented PyVision for **Qwen2.5-VL-72B** and **Qwen3-VL-235B** by providing them with a code-execution environment. As shown in updated **Table 2**, their scores improved to **24.0** and **24.9**, respectively. These results indicate that adding a runtime environment can indeed boost performance for large open MLLMs, though we observed that approximately 10% of the generated code remained non-executable.
>
> > Q1: Figure 3 is unclear and difficult to interpret. The authors should provide a more detailed explanation.
>
> **A1:** We apologize for the lack of clarity. We have significantly revised its accompanying caption in the new manuscript. We provide a detailed explanation here. In Figure 3, Function calling uses a predefined rotation function, while code writing requires the model to implement it. P1 includes only the question; P2 adds a hint to use the function; P3 explicitly instructs testing each degree in the answer choices.
>
> We highlight three key findings: (1) **Performance hierarchy:** o3 is the strongest model, with o4-mini close behind. (2) **Prompting matters:** Prompt 3 explicitly instructing models to test each degree yields the best results, suggesting models often need clear guidance to use functions effectively. (3) **More function calls in newer models:** Advanced models (e.g., o3) make significantly more function calls than earlier ones (e.g., GPT-4o), indicating improved training for iterative function use.
>
> References
>
> [1] Thinking with images. OpenAI.
>
> [2] DeepEyes: Incentivizing "Thinking with Images" via Reinforcement Learning.
>
> [3] PyVision: Agentic Vision with Dynamic Tooling

---

### Author Response · Authors · 2025-11-21
**General Response**

We sincerely thank all four reviewers for their time and valuable, constructive feedback. We are encouraged that the reviewers generally recognized the novelty and core contributions of TIR-Bench.

We summarize the recognized contributions and revisions in the updated paper here and provide separate responses to each reviewer.

# Main contributions.
- **Novelty.** The paper identifies and elevates the task of "thinking-with-images" beyond simple operations to more complex and diverse capabilities [8Z6B]; It proposes tasks that are cleverly designed to be unsolvable by static observation alone, compelling models to use a "think-tool-observe" loop [WufX]; The paper also identifies more difficult tasks suitable for new code-execution-based approaches than previous datasets [Gpt1].
 - **Analysis and Experiments.** The benchmark is comprehensive, covering 13 diverse tasks [ALVA]; The experiments clearly demonstrate that tool use is essential for these tasks and quantify the significant room for improvement, even for SOTA models [WufX]; The benchmark is noted as being sufficiently challenging [ALVA], highlights issues with current SOTA models [Gpt1], and includes a wide selection of model evaluations [Gpt1].
- **Impact.** The paper addresses a significant gap [WufX] and a critical need [Gpt1] for new benchmarks in the area of agentic or tool-assisted image reasoning; A central contribution is that the benchmark explicitly investigates how models interact with images, which is crucial for understanding agent use behavior [ALVA]; It is recognized as a potentially useful, long-term suite for measuring MLLM reasoning ability [ALVA].

# The key changes in the updated paper and more experiments and analysis included:
- We include six additional models for a broader evaluation in **Table 2**.
- We re-implemented PyVision for Qwen2.5-VL-72B and Qwen3-VL-235B by providing them with a code-execution environment and include the results in **Table 2**.
- We conducted additional experiments on agentic SFT on InternVL3-8B and Qwen-2.5-VL-7B and evaluated its performance on the rotation game and OCR tasks.
- We have significantly revised the caption of **Figure 3** to enhance clarity.
- We analyzed the error types of the PyVision model in **Appendix H**.
- We summarize the created tools of PyVision response for each task in **Apppendix I**.
- We revise **Sec. 3.2** to offer a clearer description of the annotation and checking procedures.
- We show the experimental results of synthetic data and human-annotated data on TIR-Bench on **Appendix F.**
- We add detailed sources for each task in **Appendix B** and **Section 3**.
- We have redesigned figures showing model responses with interleaved thinking/code (**Figure 4, 5, 24, 25**).
- We evaluate PyVision's efficiency by analyzing interaction turns (detailed in **Appendix J and Figure 26**).

---

### Author Response · Authors · 2025-12-02
**Summary Message to AC**

**Dear AC and SAC,**

We want to first extend our sincere appreciation for your valuable time and effort dedicated to the review process in this extraordinary situation. We wish to share that we provided a detailed rebuttal and updated paper that we believe fully addressed the concerns raised by the reviewers. As a result:

- One reviewer (**WUfX**) explicitly acknowledged that our rebuttal fully addressed their concerns and **raised their score to 6** well before the incident occurred.
- Unfortunately, the other three reviewers (with original scores **6, 4, 4**) could not respond to the rebuttal before the incident.

However, we note that **all reviewers acknowledged the importance, novelty, and contribution** of this benchmark in their initial reviews. Their specific concerns regarding the benchmark construction process, additional experiments, and visual clarity were **thoroughly discussed and addressed** in our rebuttal.

We respectfully believe this work should be judged on this positive momentum and our successful rebuttal. To aid your assessment, we provide a transparent summary of the discussion timeline and our responses below.

### **Discussion timeline**

The following events document the full discussion process. We highlight that Reviewer **WUfX** expressed their satisfaction with our rebuttal responses and raised their score prior to the incident (27 Nov, 2025):

- **21 Nov, 2025** — We submitted our full rebuttal along with the updated manuscript.
- **26 Nov, 2025** — Reviewer **WUfX** responded: *"I sincerely appreciate the authors for their response. It has addressed my concerns. I will update my rating to 6."* **(Raised score 4 → 6)**
- **26 Nov, 2025** — We thanked the reviewer for their thoughtful engagement and updated score.

### Brief contribution & rebuttal summary

**Our contributions.**

- **Novelty:** We identify and elevate **"thinking-with-images"** to complex capabilities beyond simple operations (**8Z6B**). Our tasks are cleverly designed to be unsolvable by static observation alone, compelling models to adopt a "think-tool-observe" loop (**WUfX**) and serving as a testbed for code-execution approaches (**Gpt1**).
- **Analysis and Experiments:** We present a comprehensive benchmark of 13 diverse tasks (**ALVA**) with extensive model evaluations (**Gpt1**), demonstrating that tool use is essential even for SOTA models (**WUfX**, **ALVA**). Additionally, reviewers highlighted our function calling study (**Gpt1**) and SFT comparison (**WUfX**) as interesting and insightful.
- **Impact:** We address a **significant gap and critical need** for benchmarks in agentic/tool-assisted image reasoning (**WUfX**, **Gpt1**). TIR-Bench is recognized as a useful, long-term suite for measuring MLLM reasoning ability and understanding agent interaction behavior (**ALVA**).

**Reviewers' concerns (pre-rebuttal).** To summarize, the reviewers mainly raised 4 concerns pre-rebuttal:

1. **Limited evaluation** of more open-source models and open-source models with code interpreters. (**8Z6B**, **WUfX**)
2. **Fairness of tool-use comparison** (Model with Code Interpreter vs. Static model). (**WUfX**, **ALVA**)
3. **Data scale, verification, and provenance.** (**ALVA**, **Gpt1**)
4. **Generalization** of the Agentic vs. Direct SFT pilot study. (**ALVA**, **WUfX**)

**Our responses (rebuttal).** We provided the following fixes (also reflected in the updated manuscript) to address their concerns:

- **Expanded Model Evaluation.** We added **6 additional models** (including DeepSeek-VL-2, Phi-4) and re-implemented PyVision for **Qwen2.5-VL-72B** and **Qwen3-VL-235B** with a code-execution environment, showing significant performance boosts. (Response to **8Z6B**, **WUfX**; Table 2)
- **Clarified Tool-Use & Analyzed Reasoning.** We clarified that the poor performance of static/non-agentic models serves to validate that TIR-Bench effectively targets genuine **thinking-with-images** ability. We also clarified the distinction between **tool-generation** (o3, PyVision) and **pre-defined tools** (Deepeyes). Additionally, we provided a detailed error analysis of PyVision (37% perception, 61% reasoning) and an efficiency analysis of interaction turns. (Response to **WUfX**, **ALVA**; Appendix H, I, J)
- **Strengthened Data Rigor.** We clarified the implementation of our **dual-verification strategy** during data annotation with two PhD students, explicitly defined the split between synthetic and hand-annotated tasks, and added a detailed **Data Provenance Statement** ensuring all data is permissible for use. (Response to **ALVA**, **Gpt1**; Appendix B, F)
- **Expanded SFT Pilot Study.** We conducted additional experiments on **Qwen-2.5-VL-7B** and **InternVL3-8B** for Rotation Game and OCR tasks, confirming the trend that Agentic SFT outperforms Direct SFT. (Response to **ALVA**, **WUfX**)

Thank you once again for your valuable time and effort.

---

### Meta-Review · Area_Chair_XioN · 2026-01-07

**Summary:**

This is an interesting work, but received initial reviews that were mixed but mostly negative-leaning. Post-rebuttal, there is some limited score increase (actual and projected), but ultimately still borderline regard from reviewers.

Some concerns include comprehensiveness of evaluation, small size of benchmark, tool use, and low-level nature of some tasks. The rebuttal provided additional results that mostly address the evaluation issues. The authors argue that it's a benchmark rather than a dataset, hence the size is adequate; this is only partly convincing in my view. The issue about tool use I think deserves a more comprehensive rethink. Ultimately, I think it's a nice starting point but perhaps not quite there yet. I generally find this direction interesting and promising, but perhaps further improvements (beyond the rebuttals) are needed to pass the acceptance bar for a highly-selective venue like ICLR, particularly in comparison to other submissions with stronger contributions and regard.

**Reviewer Concerns:**

Pls see above.

**Reviewer Scores:**

-- WUfX: explicitly raised to 6.
-- Gpt1 and ALVA: 50/50 chance raise to 6.
-- 8Z6B: likely maintain at 6.

---

### Decision · Program_Chairs · 2026-01-26

Reject